# Automated imaging and identification of proteoforms directly from ovarian cancer tissue

John P. McGee [1,7], Pei Su [1,7], Kenneth R. Durbin[2], Michael A. R. Hollas[2], Nicholas W. Bateman [3,4], G. Larry Maxwell[4,5], Thomas P. Conrads [4,5], Ryan T. Fellers[2], Rafael D. Melani[1], Jeannie M. Camarillo [1,6], Jared O. Kafader [1,6] & Neil L. Kelleher [1,2,6] ✉

The molecular identification of tissue proteoforms by top-down mass spectrometry (TDMS) is significantly limited by throughput and dynamic range. We introduce AutoPiMS, a single-ion MS based multiplexed workflow for top-down tandem MS (MS²) directly from tissue microenvironments in a semi-automated manner. AutoPiMS directly off human ovarian cancer sections allowed for MS² identification of 73 proteoforms up to 54 kDa at a rate of <1 min per proteoform. AutoPiMS is directly interfaced with multifaceted proteoform imaging MS data modalities for the identification of proteoform signatures in tumor and stromal regions in ovarian cancer biopsies. From a total of ~1000 proteoforms detected by region-of-interest label-free quantitation, we discover 303 differential proteoforms in stroma versus tumor from the same patient. 14 of the top proteoform signatures are corroborated by MSI at 20 micron resolution including the differential localization of methylated forms of CRIP1, indicating the importance of proteoform-enabled spatial biology in ovarian cancer.

Proteoforms containing post-translational modifications and sequence variations are the molecular products of gene expression and key effectors of biological function[1]. In addition to the molecular profile of endogenous proteoforms, complex phenotypes manifest as proteome heterogeneity across functional tissue in time and space[2]. Mass spectrometry (MS) imaging can profile proteins with moderate spatial resolution without the need for affinity reagents or antibody labels[3,4]. However, proteome complexity presents a grand challenge for "top-down" MS of intact proteoforms in tissue. When using electrospray ionization-based methods for proteoform desorption from tissues, each proteoform generates a distribution of charge states,

resulting in congested spectra in the mass-to-charge (m/z) domain. As a result, only a few of the most abundant proteoforms can be targeted in traditional data acquisition tandem MS (MS²) schemes[5].

Recently, tissue spatial profiling and imaging with intact proteoform-level information has been demonstrated using techniques such as matrix-assisted laser desorption/ionization[4,6,7], desorption electrospray ionization (DESI)[8], nanospray desorption electrospray ionization (nano-DESI)[9–12], picosecond infrared laser desorption by impulsive excitation[13], and nanodroplet processing in one pot for trace samples[14,15]. However, due to the limited pulses of ions from sampling tissue voxels versus the ion requirements for top-

[1]Departments of Molecular Biosciences, Chemistry, and the Feinberg School of Medicine, Northwestern University, Evanston, IL, USA. [2]Proteomics Center of Excellence, Evanston, IL, USA. [3]Henry M. Jackson Foundation for the Advancement of Military Medicine, Inc, Bethesda, MD, USA. [4]Department of Gynecologic Surgery and Obstetrics and the Gynecologic Cancer Center of Excellence, John P. Murtha Cancer Center, Uniformed Services University of the Health Sciences, Bethesda, MD, USA. [5]Women's Health Integrated Research Center, Inova Women's Service Line, Inova Health System, Falls Church, VA, USA. [6]Department of Biochemistry and Molecular Genetics, Feinberg School of Medicine, Northwestern University, Chicago, IL, USA. [7]These authors contributed equally: John P. McGee, Pei Su. ✉e-mail: n-kelleher@northwestern.edu

down fragmentation, molecular identification at intact level is biased to high abundance and low molecular weight proteoforms (<30 kDa)[9]. Top-down MS fragmentation on tissue has been advanced to proteoforms <70 kDa using proteoform imaging MS (PiMS)[16] by interfacing nano-DESI[9] with I²MS[17]. I²MS is an Orbitrap-based charge detection MS technique allowing single ion detection of intact proteoforms or their fragments[17] with >500× greater sensitivity and 10× higher resolving power over traditional ensemble data acquisition[18]. Herein, we developed "AutoPiMS", a PiMS-derived[16] data-dependent MS² workflow[9,19] for multiplexed proteoform identification directly off tissues and applied it to study spatial biology in human ovarian cancer. AutoPiMS augments PiMS with a computational engine for unattended proteoform target selection and acquisition method generation, and is directly interfaced with high-throughput data processing and database search. AutoPiMS streamlines multiplexed on-tissue top down proteomics and can be readily interfaced with a variety of electrospray-based protein MS imaging modalities to extend proteome coverage in spatial proteomics, advancing the field of molecular histology.

## Results and discussion
### Overview of the AutoPiMS workflow
AutoPiMS achieves unattended identification of proteoforms using a semi-automated spatially-aware, data-dependent acquisition strategy. The detailed logic of the AutoPiMS workflow is depicted in Fig. 1a: step (1) a PiMS line scan to obtain proteoform absolute mass, charge, and spatial distribution (Fig. 1a, top left); step (2) automated or manual target selection using an algorithm that finds optimal isolation windows for targeted fragmentation; and step (3) automated creation of an MS² acquisition method within the contexts of $m/z$ and tissue location. MS² fragmentation is then performed by running the PiMS probe across a fresh line parallel to the survey line scan but offset by ~20 μm (Fig. 1a, top right). MS² fragmentation data were acquired in an unattended manner in either conventional ensemble mode for proteoform targets <17 kDa at 14 Hz or I²MS mode[16,17] for proteoforms >17 kDa at 1 Hz (Methods). The workflow includes a search engine[20,21] for proteoform identification compatible with the individual ion, MS² data type (Fig. 1a, bottom)[17]. All steps aside from data transfer and MS² method setup are fully automated and can be customized manually as desired. We note that the first step in the AutoPiMS workflow is not limited to a single line scan but can also be applied to a PiMS imaging experiment with space between lines reserved for MS² data acquisition. Direct infusion of a standard mixture of six intact proteins was employed for proof-of-concept, and the workflow readily characterized all six components at the MS² level (Methods and Supplementary Fig. 1).

### AutoPiMS applied to HGSOC tissue
The AutoPiMS workflow was applied to a 10 μm thin section of fresh-frozen human high-grade serous ovarian cancer (HGSOC) tissue (~95% tumor cellularity, Fig. 1a, very top). We obtained 113 proteoform masses at >1% relative abundance ranging from 4–67 kDa from a single 40-min survey line scan (Methods, Supplementary Fig. 2, and Supplementary Table 1 within Supplementary Data 1), of which 25 were in the 17–50 kDa mass range (mass spectrum shown in Fig. 1b). The reconstructed survey scan shown in Supplementary Fig. 3 illustrates the high spectral complexity in the $m/z$ domain, making it challenging to isolate and target these proteoforms.

Next, the AutoPiMS workflow engaged its algorithm (Methods) to direct data acquisition toward proteoform targets across both the $m/z$ and spatial dimensions (where the target is at or near its highest abundance). The algorithm identified favorable $m/z$ windows for 87 of the 113 proteoform masses (Supplementary Table 2 within Supplementary Data 1). Larger proteoforms are challenging targets, as their signals are diluted into many charge state and isotopic channels[5]. Additionally, their signals often overlap with small proteoforms in the

$m/z$ domain (Supplementary Fig. 4). However, the process was able to identify favorable $m/z$ windows for all 25 targets in the 17-50 kDa mass range (Fig. 1c, Supplementary Table 2 within Supplementary Data 1, and Supplementary Fig. 5) at or near their top abundance in space (Fig. 1d), even if they would have gone unannotated in a traditional $m/z$ data-dependent acquisition mode. Combined with the optimized spatial locations for these 25 targets, tandem MS data acquisition in I²MS mode and database searching identified 23 of the 25 targets (Fig. 1b, Supplementary Table 3 within Supplementary Data 1, and Supplementary Data 2), with the quality of proteoform information being at Level 2 A or better using the five-level proteoform classification system[22]. The same AutoPiMS workflow was repeated on two adjacent 10 μm sections of the tumor from the same patient, identifying 69% (18 out of 26) and 80% (16 out of 20) of the AutoPiMS-selected targets achieving an averaged success rate of >80% (Supplementary Tables 3–5 within Supplementary Data 1). In addition, the AutoPiMS workflow allowed spatially-enhanced targeting of larger proteoforms. In Fig. 1e, we show the mass spectrum and spatial distributions of two ~53.6 kDa proteoforms in a survey line scan from another HGSOC tissue specimen. The workflow generated MS² spectra containing sequence tags for identification using ≤100 MS² scans (Fig. 1e, bottom left and right).

For proteoforms with masses <17 kDa, we found that MS² in ensemble mode provides comparable information to MS² in I²MS mode (Methods). To achieve a higher data acquisition rate for targets in <17 kDa range, we performed AutoPiMS workflow using Orbitrap Exploris 480 implemented with a 4 kV central electrode[23]. Using AutoPiMS-embedded algorithms, we obtained favorable isolation conditions for 79 targets detected in the <17 kDa range in a representative survey line scan on the HGSOC tissue described in Fig. 1a (Supplementary Fig. 6). Subsequent MS² acquisition found matching candidates for 40 proteoforms above 1% relative abundance passing either a 1% false discovery rate (FDR) or manual inspection (Supplementary Table 6 within Supplementary Data 1). From the above-mentioned set of three runs for proteoforms >17 kDa and three runs for proteoforms <17 kDa (Supplementary Tables 6–8 within Supplementary Data 1) on adjacent tissue sections from the same patient, the workflow identified a total of 73 proteoforms at the MS² level (Supplementary Table 9 within Supplementary Data 1). A gene ontology analysis using the 73 proteoforms showed that *<cellular responses to stress>*, *<ribosome, cytoplasmic>*, and *<metalloprotease deubiquitinases>* were the highest enriched terms in HGSOC with *p values* of $10^{-18}$, $10^{-12}$, and $10^{-6}$, respectively (Fig. 1b and Supplementary Fig. 7).

### Spatial proteoform biology of complex HGSOC tissue
We deployed AutoPiMS to identify proteoform signatures in histology-defined tumor versus stromal regions within HGSOC tissue from a single patient[24]. Cancer-associated stroma co-mingles with cancer cells in tumor tissues in complex spatial arrangements and plays important roles in tumor cell invasion and metastasis[24]. Proteoform signatures enriched in tumor versus stroma were first detected using label-free quantitation (LFQ, Fig. 2a). In the LFQ, 240 sampled regions were selected from either tumor or stromal regions on the same section, and ~50 I²MS scans were obtained for each sampled region (Fig. 2a and Methods). Total ion count distributions showed overall lower protein abundance in stroma (Supplementary Fig. 8). A total of 1013 proteoform masses were detected, and 552 proteoforms showed statistically significant differential ion counts between tumor and stroma regions filtered by a conservative 1% FDR (Supplementary Table 10 within Supplementary Data 1)[25]. The tumor and stroma sampled regions were subjected to principal component analysis using the ion counts of the 552 proteoforms and were readily separated into distinct clusters (Fig. 2d). The statistical test and unsupervised classification were shown to be reproducible in two additional LFQ technical replicates obtained from adjacent tissue thin sections from the same HGSOC

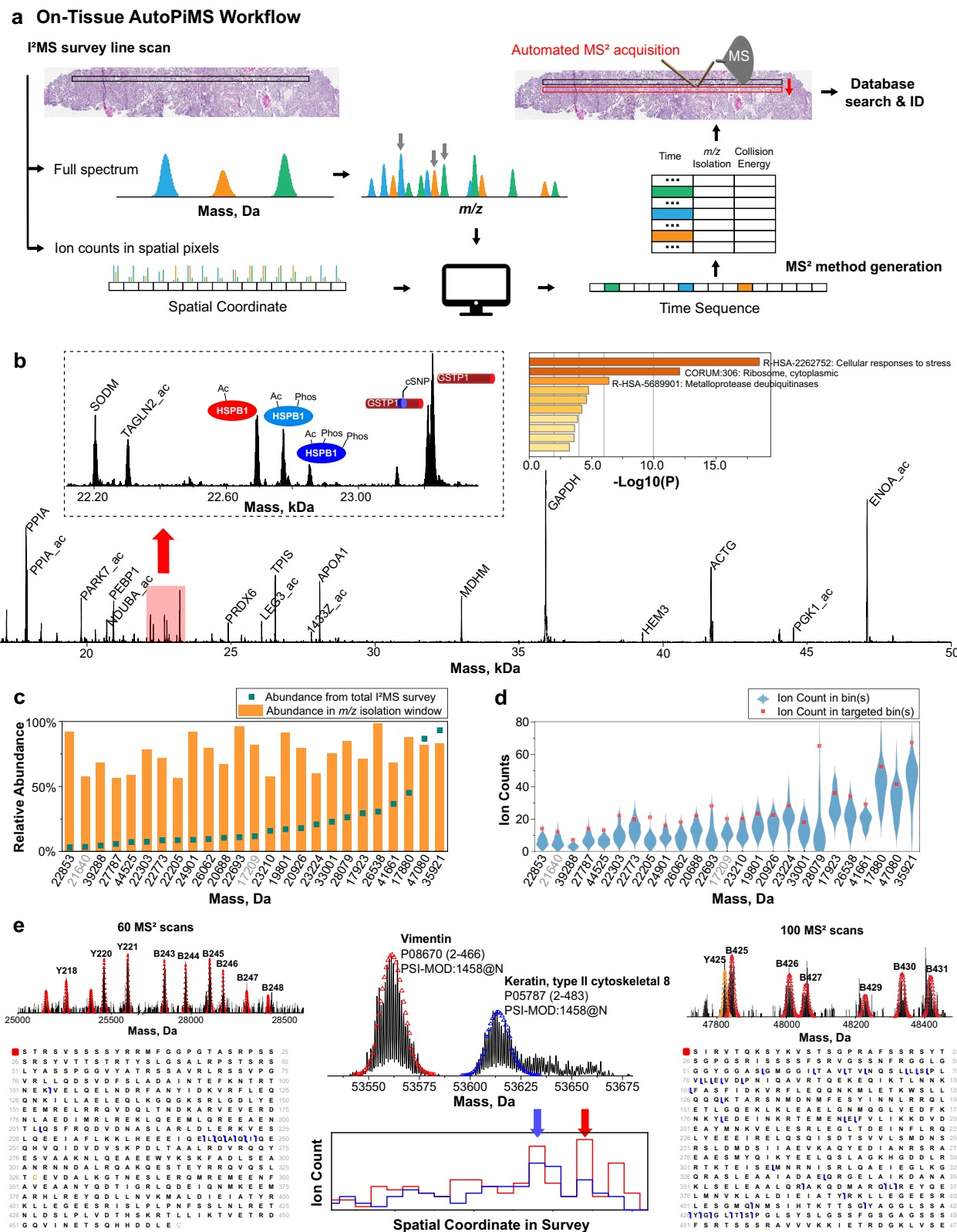

tumor tissue specimen (Supplementary Tables 11 and 12 within Supplementary Data 1, and Supplementary Fig. 9). The LFQ analysis identified 303 differentially abundant proteoforms between the tumor and stromal regions with |log₂(fold change)| > 0.5 (Fig. 2b, Supplementary Table 13 within Supplementary Data 1). The 303 proteoforms were annotated using intact mass tag search against a custom database constructed from proteins found by the Clinical Proteomic Tumor

Analysis Consortium (CPTAC) analyses of ovarian cancer (Methods). Using a stringent 1.5 ppm mass tolerance, 114 proteoforms could be putatively identified using their intact mass from a set of ovarian cancer proteins (Supplementary Table 13 within Supplementary Data 1).

In the next step, we compared the LFQ results with a PiMS imaging experiment performed on a region with spatially comingled tumor and

**Fig. 1 | Logic and performance metrics of AutoPiMS proteoform identification directly from ovarian cancer tissue. a** A survey line scan produces individual ion mass spectra detecting multiply-charged proteoform ions under denaturing conditions. Selected charge states of proteoforms are automatically targeted at optimized locations in a subsequent line scan for top-down fragmentation and database search. **b** A survey spectrum in the 17–50 kDa range with proteoform identification by the AutoPiMS workflow. The left inset shows a zoomed view of the spectral region highlighted in red. In this region, we detected phosphorylated proteoforms of heat shock protein beta-1 (HSPB1, UniProt accession: P04792) and a coding polymorphism that created two proteoforms of glutathione S-transferase P (GSTP1, UniProt accession: P09211). The right inset shows gene ontology analysis of 73 MS[2]-identified proteoforms. **c** A bar plot showing the difference in abundance of 25 specific proteoforms in the survey line scan (teal dots) versus when they were fragmented in the subsequent identification scans (orange bars). **d** Raw ion counts of the 25 proteoforms in the survey scan when spatial bins are assigned to the 25 targets randomly (blue violin plots) versus algorithm-optimized (red dots) in subsequent MS[2] fragmentation scans. **e** MS[1] survey (middle) and MS[2] spectra (outer) along with graphical fragment maps of N-terminal acetylated vimentin (UniProt accession: P08670, left) and N-terminal acetylated keratin type II cytoskeletal 8 (UniProt accession: P05787, right) identified by AutoPiMS. In the middle panel, theoretical isotopic distributions of the two proteoforms are overlayed. The step plots (middle, bottom) show the spatial distributions of the two proteoforms along the survey line and the locations where they were targeted (vimentin in red, keratin in blue). Source data are provided as a Source Data file for (**c**–**e**).

stromal compositions (Fig. 2a). In this imaging dataset, 618 proteoforms were detected above 0.1% relative abundance (Supplementary Table 14 within Supplementary Data 1, and Supplementary Fig. 10). 249 common proteoforms were found in both the imaging and the LFQ dataset, featuring a 45% overlap in proteome coverage. Figure 2b shows PiMS images of 17 of the proteoforms with highly differential ion counts between tumor and stromal regions in LFQ. Distinct tumor or stromal localization was observed in these 17 images and was consistent with the tumor or stromal enrichment of the corresponding proteoform found in LFQ. Moreover, these results were consistent in z-stack among three technical replicates obtained in same regions across three adjacent 10 μm sections (Fig. 2c).

AutoPiMS was subsequently employed for direct top-down MS[2] of these proteoform signatures. A survey line scan followed by an adjacent MS[2] scan was performed along the dashed line in Fig. 2a next to the LFQ-profiled tumor and stromal regions. All 17 proteoforms were detected in the survey scan, 16 of them were MS[1] annotated, and 14 of them were MS[2] identified (Supplementary Table 15 within Supplementary Data 1). In Fig. 2e, we show the precise identification of tropomyosin alpha chain isoforms with >95% identical sequences known to exhibit fibroblast-specific expression (Supplementary Fig. 10)[26]. The localization of these tropomyosin isoforms in imaging is consistent with fibroblasts being a major stromal cell type in the imaged region (Fig. 2b, e). This result is also consistent with previously published tumor- and stroma-enriched bottom-up proteomics studies on HGSOC biopsies (Supplementary Fig. 11)[24]. We note that the method generation step in the AutoPiMS workflow may be paired with the LFQ output to selectively target proteoforms with the highest fold changes and highest confidences (Methods) and may be interfaced with imaging dataset when proteoform image classifiers are being developed in the future.

In the imaging experiment described above, methylated proteoforms of the protein CRIP1 were all detected at significantly higher levels in tumor regions and showed similar spatial distributions (Fig. 2b, c). The MS[2] data support the placement of the mono- and dimethylation on Arg68 as a major modification site (Fig. 2f, lower right, and Supplementary Fig. 12). Monomethylated CRIP1 is observed as the dominant CRIP1 proteoform in a majority of the locations on tissue[27]. Notably, when imaging experiment was performed on an adjacent region of the tissue (Fig. 2f, left), these three proteoforms showed significantly different spatial distribution in tumor and stromal regions. In particular, some vascularized locations in stroma showed higher relative levels of unmethylated CRIP1, whereas these regions excluded the Arg68me2 form of CRIP1 (Fig. 2f, right). Vascularized regions were asserted using microscopy images and co-localization of protein markers of cells comprising capillaries like vimentin (Fig. 2f, left)[28]. This explains the highly variable CRIP1 abundances in our previous bottom-up proteomics study on tumor- or stroma-enriched homogenized HGSOC samples (Supplementary Fig. 11)[24]. Given the ~20 microns lateral spatial resolution and 80 microns line spacing, cell-specific observations as well as the functional role of Arg68me0 in angiogenesis will require future study (e.g., with proteoform-specific

affinity reagents and light microscopy). Previous studies have reported elevated level of CRIP1 expression in gastric, prostate, and ovarian cancers[29–31]. The higher level of dimethylation observed in large tumor regions may be attributed to PRMT overexpression[32] or mutation-induced PRMT enzymatic activity. However, RNA-seq data from this patient does not show overexpression or mutation for any member of the PRMT gene family member[33].

In summary, we developed an integrated platform, AutoPiMS, to drive future advancements in proteoform-level spatial biology. The platform enables four-dimensional characterization of proteoform signatures: intact molecular mass, spatial distribution, quantitative analysis of differential expression, and molecular identification. In particular, the automated data acquisition engine enables proteoform identification up to ~54 kDa at a speed of <1 min per proteoform directly off tissue. We found ~300 proteoforms differentially detected in tumor versus stromal regions from ovarian cancer sections, where the patient can serve as their own control depending on the profiled regions in the experiment. This platform fills the gap between high-confidence proteoform discovery and spatial proteomics, opening up a new avenue for discoveries and precision diagnostics in clinical histology.

## Methods
### Ethical statement
This research complies with all relevant regulations. All study protocols were approved for use under a Western IRB-approved protocol, "An Integrated Molecular Analysis of Endometrial and Ovarian Cancer to Identify and Validate Clinically Informative Biomarkers" deemed exempt under US Federal regulation 45 CFR 46.102(f). All experimental protocols involving human data in this study were in accordance with the Declaration of Helsinki and informed consent was obtained from all patients. Patient information was deidentified and the researchers were blinded to any individual-level data information in this study.

### Tissue/sample preparation
All materials used are commercially available and listed in the Methods section. Optimal cutting temperature-embedded HGSOC tissues were cryo-sectioned (10 μm), thaw mounted onto glass microscope slides (Indium tin-oxide coated glass, Delta Technologies, CG-81IN-S115, Fisher Scientific, Waltham, MA) and stored at −80 °C before MS analysis. Detailed information of the tissue sections (H&E stained images, percentage of tumor, and tissue sites) used in this study are included in Supplementary Fig. 13.

The HGSOC tissue sections were thawed at room temperature in a desiccator under slight vacuum, fixed and desalted via successive immersion in 70%/30%, 90%/10%, and 100%/0% (v/v) ethanol/water solutions for 20 s each, delipidated by 99.8% chloroform for 60 s, and dried under slight vacuum. The tissue sections were scanned using a PathScan Enabler (Fisher Scientific, Waltham, MA) prior to experiments.

Pierce Intact Protein Standard Mix was purchased from Fisher Scientific (Waltham, MA) and diluted in LC-MS grade 60%/39.5%/0.5%

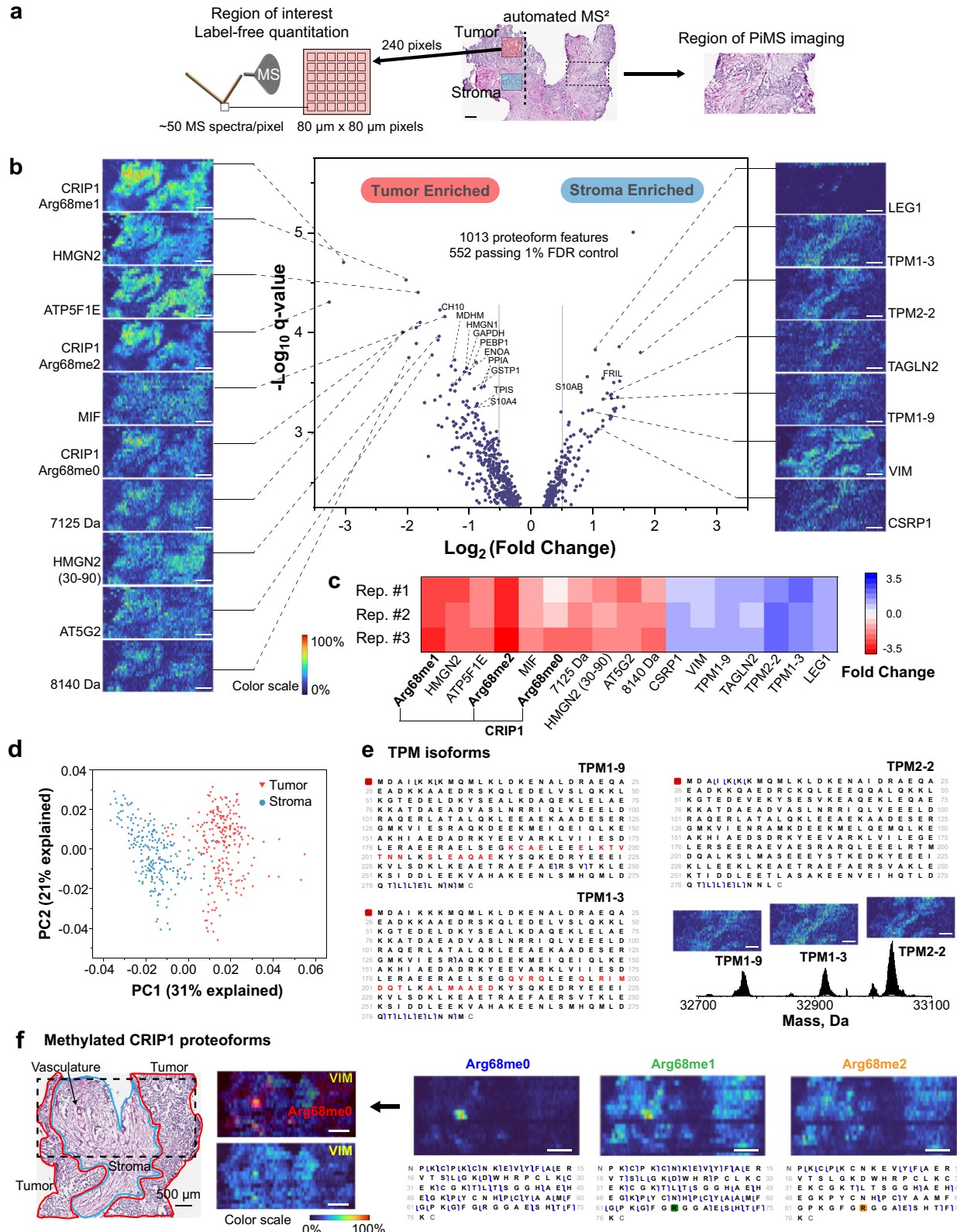

**Nature Communications** | (2023)14:6478

($v/v/v$) water/acetonitrile/glacial acetic acid solvent to make a 1 ng/µL solution for direct infusion.

**PiMS ion source and sampling conditions**

Proteoform imaging mass spectrometry (PiMS) utilizes nano-DESI MS for spatial profiling of thin tissue sections, which has been described in detail elsewhere[34,35]. Briefly, the nano-DESI probe is comprised of a primary capillary to deliver extraction solvent to the tissue and a nanospray capillary for analyte transfer and ionization at MS[35]. Localized analyte extraction from tissue is sampled by a dynamic liquid bridge formed between the junction of the two capillaries and the tissue section. While moving a sample under the PiMS probe in line motion, analytes from different spatial locations were sampled as individual ions and transferred to MS for detection[36]. Typical scan

**Fig. 2 | Applying AutoPiMS to ovarian cancer tissue, including PiMS imaging, label-free quantitation, and automated MS² identification. a** Regions of interest used for label-free quantitation (left) and imaging (right). Dashed line in the middle image depicts an AutoPiMS line scan. **b** Volcano plot (middle) generated from label-free quantitation of 552 proteoforms using the ion counts from 472 sampled regions in tumor and stroma. PiMS images of 10 proteoforms significantly enriched in tumor (left) and 7 proteoforms in stroma region (right) are correlated to their quantitation outcome in the volcano plot by dashed lines. Proteoforms labeled in the volcano plot are MS¹ annotated. **c** Reproducibility of quantitation for the 17 differentially detected proteoforms highlighted in (**b**). **d** Principal component analysis of 472 region-of-interest samples (using the ion counts of 552 proteoforms as dimensions) showing clear differentiation of tumor vs. stroma samples (red and blue, respectively). **e** Examples of the identification of highly-similar proteoforms,

tropomyosin alpha-1 chain isoform 9 (UniProt accession: P09493-9), tropomyosin alpha-1 chain isoform 3 (UniProt accession: P09493-3), and tropomyosin beta chain isoform 2 (UniProt accession: P07951-2). **f** Differential spatial localization of three proteoforms of CRIP1 (UniProt accession: P50238) with Arg68me0, me1 and me2; The imaged area highlighted using a dashed box on the histology image (left) contains tumor (red), stroma (blue), and vasculature regions adjacent to the region in (**a**). PiMS images of CRIP1 proteoforms are shown at right. The graphical fragment maps obtained from automated MS² characterize unmethylated CRIP1 and localize the mono- and di-methylations of CRIP1. A PiMS image of N-terminally acetylated vimentin (UniProt accession: P08670) serves as a marker for stromal and vascularized tumor regions (middle bottom). Merged image of vimentin and CRIP1 Arg68me0 show co-localization in tumor vascular regions (middle top). All scale bars are 500 μm.

rastering rate used for different experimental workflows in this work are: 5 μm/s (PiMS imaging), 4 μm/s (LFQ), and 2–4 μm/s (AutoPiMS).

The PiMS probe used in this study was fabricated using fused silica capillaries (Molex, Thief River Falls, MN, OD/ID 150/75 μm and OD/ID 100/40 μm). To improve the spatial resolution of sampling, the primary capillary was flame-pulled to an OD of 20 μm on one side to make contact with the nanospray capillary (OD 100 μm, ID 40 μm). The width of the liquid junction formed by this probe was ~80 μm in all experiments estimated by measuring the width of the trace left by the probe on tissue using optical microscopy. To improve the stability of the liquid bridge during spatial profiling, we first measured the surface tilting angle of the tissue section by defining a three-point plane on tissue prior to experiments. All samples were electrosprayed in positive ionization mode through the PiMS probe under denaturing conditions in a 60%/39.4%/0.6% (*v/v/v*) acetonitrile/water/glacial acetic acid solution at a flow rate of 300–400 nL/s.

## PiMS imaging data acquisition and processing

PiMS experiments were conducted on the Orbitrap Exploris 480 mass spectrometer (Thermo Fisher Scientific, Bremen, German) in the I²MS mode at a resolution of 120000 at *m/z* 200 corresponding to 0.5 s Orbitrap detection period[23]. The Orbitrap mass analyzer on Exploris 480 system operates at a central electrode voltage of 4 kV, allowing for more favorable ion lifetimes for I²MS over models that operate at 5 kV. The source conditions on the mass spectrometers were set as follows: ESI voltage: 3 kV; in-source CID: 5 eV; S-Lens/Funnel RF level: 70%; capillary temperature: 325 °C. MS injection time was kept at 0.4 ms all time.

HCD pressure level was optimized on these instruments to reduce collision-induced ion decay within the Orbitrap analyzer without substantial losses in trapping efficiency in the C-trap[36]. In particular, the HCD pressure setting was kept at 0.33 (arbitrary unit) for Orbitrap Exploris 480 mass spectrometer (UHV pressure $<5 \times 10^{-11}$ Torr). 0.5–1 V extended trapping was used on the Orbitrap Exploris 480 mass spectrometer to enhance the trapping efficiency of large protein ions. Additional relevant data acquisition parameters were adjusted as follows: mass range: 400–2500 *m/z*; AGC mode: fixed; enhanced Fourier transform: off; Emeter averaging: 0; microscans: 1.

PiMS data acquisition was performed in the I²MS mode described in detail elsewhere[16]. In short, individual ions on tissue were collected by the PiMS probe as a function of locations; time-domain data files were acquired and recorded as Selective Temporal Overview of Resonant Ions (STORI) files[37]. The STORI slope of an individual ion was compared to a charge calibration curve of the instrument, and an integer charge number (*z*) was assigned to the ion by statistically evaluation using an iterative voting algorithm. The neutral mass of the ion is calculated by:

$$Mass = \left(\frac{m}{z} \times z\right) - (z \times M_{proton}) \qquad (1)$$

The neutral centroid masses of the individual ions were converted to profiles using a Kernel density estimation approach, and ions from the entire imaged area were used to construct a mass-domain spectrum. Proteoform features were identified in the spectrum using an iterative peak picking algorithm using a user-defined threshold and confirmed by a modified Thorough High Resolution Analysis of Spectra by Horn (THRASH) algorithm[38].

The PiMS imaging experiment shown in Fig. 2 was performed at a probe line scan rate of 5 μm/s and a data acquisition rate of 2 spectra/s with a strip step of 80 μm between the adjacent lines. The dimensions of the two imaged regions in Fig. 2 were 3.4 mm × 1.6 mm and were both composed of 20 line scans. PiMS proteoform images were generated using ions within the isotopic distributions of the proteoforms using a MATLAB script developed in-house. Major isotopic masses (5–11 isotopes depending on the proteoform mass) of a proteoform was first identified from the mass spectrum. A ± 10 ppm mass tolerance was used to find individual ions within the major isotopic masses composing the proteoform. The spatial locations of these ions were used to generate a two-dimensional heatmap as the proteoform image[16]. The spatial resolution of the imaging experiments were ~20 μm (Supplementary Fig. 14).

## Label-free quantitation (LFQ)

Region-of-interest spatial profiling for LFQ was performed on the Orbitrap Exploris 480 mass spectrometer (Thermo Fisher Scientific, Bremen, German) using an identical set of parameters as the PiMS imaging experiments described above except for the following parameters: probe scan rate (4 μm/s) and MS injection time (0.2 ms). The experiments were performed at a probe scan rate of 4 μm/s and a spectral acquisition rate of 2 spectra/s with a strip step of 100 μm between adjacent lines. 100 μm strip step was employed instead of 80 μm to avoid repeated sampling of the same regions, which is detrimental to quantitation.

In the LFQ experiments, we analyzed 240, 80 μm × 80 μm areas each in selected tumor and stroma regions of interest from the biopsy annotated by a histologist. In contrast to conventional LFQ sample preparation, each area was considered as an LFQ "sampled region". In LFQ experimental design, we intended to extract all proteoform signals from each sampled region without signal suppression caused by saturation. Supplementary Fig. 15 shows typical imaging profile at 5 μm/s (a) and LFQ profile at 4 μm/s (b). We did not observe signal saturation in the scans in imaging experiments, in which a higher protein concentration was analyzed (each unit volume of solvent extracts 20% more tissue area, and a 0.4 ms injection time was used rather than 0.2 ms in LFQ). Therefore, signal saturation rarely took place in LFQ experiments from these tissues.

Another important consideration for quantitation of each "sampled region" is to minimize the carryover of proteins from the previous "sampled region". For LFQ experiments for this set of cancer tissues, we optimized the rastering rate of the probe to make sure that proteins are maximally extracted in every 80 μm × 80 μm area with minimal

carryover from the adjacent areas. As demonstrated in Supplementary Fig. 15, in contrast to even sampling profiles in imaging, LFQ profile shows many spike features with highest ion counts (at the tip of the spike) ~10% to 20% higher than the average profile in imaging experiment. These profiles correspond to an almost complete extraction of protein content within every 20 μm of lateral distance. In this regard, when protein signals within 80 μm distance are binned to construct a sampled region for LFQ quantitation, each region will contain minimal contribution from adjacent 80 μm area. Another reason we chose 80 μm as the lateral size of sample is based on the width of the liquid bridge (80 μm), which eliminates the difference between lateral or horizontal orientation and mimic the process in laser capture micro-dissection, micro-punch, and other microsampling approaches.

Groups of 45–50 adjacent MS scans were binned to construct the proteoform profile of a sampled region, from which the proteoform ion counts were extracted. Charge assignment of the ions was conducted by co-evaluating all the 240 tumor and stroma LFQ sampled regions together to maintain the unsupervised nature of the analysis. Proteoform feature extraction from the sum of 480 samples >0.1% relative abundance results in similar number of fractures compared to that obtained using THRASH algorithm[38]. Due to current limitations in relative quantitation, THRASH was used only for validation but not involved in the LFQ process. Using only major isotopes for quantitation guarantees the low-interference from adjacent overlapping proteoforms in the spectrum. All other I²MS data acquisition and analysis not specific to the LFQ experiments can be found in the "*PiMS imaging data acquisition and processing*" section.

Before statistical analysis, a quality control step was performed to eliminate sampled regions with ion counts below a certain threshold (considered as empty sampled regions due to probe lost contact or localized sample contamination). In this study, sampled regions with <1000 total ions were tossed out. From this step, 237 tumor and 235 stroma sampled regions (out of 240) passed the control and were used for further analysis. In another two technical replicates, 235/237 and 227/228 tumor/stroma sampled regions passed for further analysis.

Next, for each proteoform feature, a T-test was performed between the tumor and stroma sampled regions, from which a *p*-score was obtained. The *p*-scores are converted to FDR-controlled *q*-values using the Benjamini-Hochberg (B-H) procedure[7]. In particular, we first ranked the proteoforms according to $-\log_{10}(p\text{-score})$ in a descending order, and calculated the B-H critical scores (*q*-values) using *Eq.* (2):

$$\text{B} - \text{H critical value} = -\log_{10}\left(\frac{i{*}Q}{m}\right) \quad (2)$$

where $i$ = ranking of the *p*-score, $Q$ = user-defined FDR, and $m$ = total number of sampled regions in tumor/stroma. All proteoforms with $-\log_{10}(p\text{-score})$ higher than the critical value were considered statistically significant. In this study, we calculated the critical value using 1% FDR, which resulted in 552 out of the 1013 proteoforms passing the test. In another two technical replicates, we found 616 out of 939 and 597 out of 954 proteoforms to be statistically significant under same conditions. Fold change of each proteoform was calculated using the mean ion count of that proteoform across all stroma sampled regions divided by the mean ion count across all tumor sampled regions. A volcano plot in Fig. 2b was constructed by plotting $-\log_{10}(q\text{-values})$ of the statistically significant differential 552 proteoforms against $\log_2(\text{fold change})$.

In the next step, unsupervised linear dimension reduction using principal component analysis (PCA) was performed using statistically significant proteoforms. In this workflow, PCA was employed to validate the clustering of tumor and stromal sampled regions. The workflow may be further adapted into studies with more than two biological sample. The proteoform ion counts were normalized to total ion count

from a sample and subjected to PCA using a MATLAB script built in-house. The first two columns of the *score* table were extracted to generate a scatter plot displaying the separation of tumor and stroma clusters using the first two principal components. In addition, the MATLAB script built for LFQ is bridged to automated MS² acquisition in AutoPiMS described in the section below. In particular, the target list in automated MS² can be constructed from the most significant features in the volcano plot obtained from LFQ.

## AutoPiMS algorithms

**Survey scan and target selection algorithm.** Individual ion MS (I²MS) can resolve overlapping *m/z* components in the mass domain and detect species that would otherwise disappear into the proteinaceous signal baseline[17]. However, this potential is realized as a result of data processing after acquisition and not in real time during acquisition. This combined with the fact that the measurement involves individual ions instead of clearly defined ensemble regime charge state distributions indicates that traditional data-dependent acquisition approaches are insufficient. Therefore, a precursor selection algorithm that capitalizes off of the resolution and sensitivity of the processed I²MS data is necessary.

First, a survey (*i.e.*, intact) I²MS experiment is conducted on the sample (or, in the case of tissue imaging, a line scan parallel and adjacent to the line scan to undergo fragmentation) as described previously. After processing the data via STORIBoard (Proteinaceous, Evanston, IL), the table of ions that constitute the resultant mass spectrum can be exported for further analysis.

The ion table is then processed by a fragmentation selection program (written in MATLAB in-house). A peak-picking algorithm is run through the I²MS mass spectrum to select proteoform precursors above a user-defined relative abundance threshold. These precursors can be filtered by the user to only target proteoforms of interest. Then, with the list of precursor masses, the algorithm assigns each ion to one of the target precursor masses, if possible. Specifically, if an ion's *m/z* measurement falls within the inclusion window for a given precursor mass (*i.e.*, within $\pm n$ Da of a mass identified by the peak-picking algorithm), the ion is assigned to that precursor mass. With these assignments, it is possible to reconstruct the *m/z* spectrum of each precursor, purified in silico.

After reconstructing the *m/z* spectra of all targets, the algorithm goes target by target, comparing the number of target ions in a given *m/z* window versus the number of other ions within that same window. Isolation windows for fragmentation for each species are selected on the basis of both target abundance and target window purity (*i.e.*, the limited presence of co-isolating species). Then, during any subsequent and comparable experiments, the user can "blindly" (as in, not verifying with online, real-time MS¹ information) fragment at the noted *m/z* to characterize the target. Furthermore, the algorithm notes the abundance and mass identity of co-isolating species within each target *m/z* window, which allows the user to better account for fragmentation peaks that were not annotated as originating from the intended target.

**Spatial bin allocation algorithm.** For samples that are homogenous across the entire experiment (*e.g.*, direct infusion with a syringe pump), the target selection algorithm described above is sufficient, as it outputs a prioritized list of *m/z* regions for characterization. However, in the AutoPiMS workflow on dynamic samples such as tissue, the proteoform composition and concentration changes as the sampling probe moves to different locations in the tissue being sampled. Therefore, a complementary algorithm is required to prioritize the temporal (and, therefore, spatial) order in which target *m/z* windows are characterized.

The line region is divided into a number of equally-sized bins depending on the number of targets. By default, one bin is assigned one target. The algorithm first imports the reconstructed spatial

profiles of the targets in the selected *m/z* windows identified in the previous section. The reconstructed spatial profiles detail ion counts corresponding to MS scans. MS scan indices are converted into spatial coordinates calculated from the constant, lateral motion of the PiMS probe and the MS data acquisition rate. Next, low-resolution spatial profiles are constructed to match the number of bins of the line region. This is achieved by combining ion counts from scans located within the same bin.

Next, the algorithm finds the bin containing the highest ion counts for each target as its desired spatial assignment for MS². Some bins may not be the highest ion count for any of the targets, which, therefore, are left unassigned. Bins containing a unique target are immediately assigned to the target. Further optimizations are performed by the algorithm for bins ideal for more than one target. In particular, the bin is prioritized and assigned to the target with the lowest absolute ion count among all the targets contained within this bin. All the rest of the targets that do not find a bin are returned to the pool for the next round to look for secondary bin options. In this round, the algorithm finds the second highest ion count for the unassigned targets within the rest of the bin indices and repeats the same decision-making process as described above. The algorithm iterates the decision-making process until all targets are assigned a unique bin. For the current experimental scale (*i.e.*, containing up to ~100 targets), the algorithm reaches convergence within ten iterations.

The algorithm is also able to accommodate MS² in I²MS mode for fewer targets, in which multiple bins may be assigned to one target. This allows for targeting >17 kDa proteoforms using MS² in I²MS mode, which requires substantially more MS² scans to build ion statistics for fragment matching than proteoforms <17 kDa. In this version of the algorithm, the line region is divided into 150–300 μm bins to represent the spatial variations of the proteoforms without sacrificing the targets with most desirable locations lumped in a large bin. For the current scale of the experiment, up to two bins may be assigned to each target. Using the same logic, the algorithm finds the most favorable bin for each target in the first step. The second step assigns one of the rest of the bins to targets of the lowest total abundances following the same logic. In addition, the algorithm contains an add-in function to adjust the number of bins for maximizing the prioritization of large proteoforms at relatively low abundances.

## AutoPiMS data acquisition

**MS/MS of direct infusion standard mixture.** Pierce Intact Protein Standard Mix (Thermo Fisher Scientific, Waltham, MA) was introduced into the mass spectrometer via a HESI source at a flow rate of 1.5–2 μL/min. MS¹-level data acquisition for Pierce Intact Protein Standard Mix was conducted either on a Q-Exactive Plus or a Q-Exactive HF mass spectrometer (Thermo Fisher Scientific, Bremen, Germany). On both Q-Exactive Plus and Q-Exactive HF mass spectrometers, the Orbitrap central electrode voltage was adjusted to 1 kV on the to improve ion survival rate and resulting individual ion charge assignment[36]. The source conditions on the mass spectrometers were set as follows: ESI voltage: 3 kV; in-source CID: 0–15 eV; S-Lens/Funnel RF level: 70%; capillary temperature: 325 °C. Injection time for Pierce Intact Protein Standard Mix survey was fixed at 0.1 ms. The HCD pressure setting was kept at 0.5 (arb.) on both mass spectrometers. Additional relevant data acquisition parameters were adjusted as follows: mass range: 400–2500 *m/z*; AGC mode: fixed; enhanced Fourier transform: off; Emeter averaging: 0; microscans: 1. MS² experiments for direct infusion of the Pierce Intact Protein Standard Mix targets were performed using higher-energy collisional dissociation (HCD) with an injection time of 1–60 ms. The MS/MS precursor isolation windows for the six proteoform targets were manually defined in "AIF" mode with HCD collision energies ranging from 8 to 12 eV. Additional parameters in MS² in I²MS mode were set as follows: Mass range: 200–2000 *m/z*; AGC mode: fixed; enhanced Fourier Transform enable: Off.

**General schematic AutoPiMS workflow.** Schematic AutoPiMS workflow and typical time in each step are listed in Supplementary Fig. 16. Aside from what has been described in the main text, we note that the third step ("identify targets"), a Method.csv file is generated, which is subsequently imported into a method file (XCalibur 3.0 user interface on Orbitrap Exploris 480 is used as an example in the screenshot).

**MS/MS of <17 kDa proteoforms on tissue.** Data acquisition for fragmentation-based analysis of <17 kDa proteoforms was performed on an Orbitrap Exploris 480 mass spectrometer (Thermo Fisher Scientific, Bremen, Germany). The MS¹ survey line scan in I²MS mode was performed using identical parameters as the PiMS imaging experiments described in the "*PiMS imaging data acquisition and processing*" section at a probe scan rate of 2–4 μm/s. In particular, MS injection time was set 0.2 ms, and the MS resolution was set at 120000 at *m/z* 200 corresponding to 0.5 s Orbitrap detection period. The HCD pressure setting was kept at 0.33 (arb., UHV pressure <5 × 10⁻¹¹ Torr). Extended trapping was kept at 0.5–1 V was used on the Orbitrap Exploris 480 mass spectrometer to enhance the trapping efficiency. Additional relevant data acquisition parameters were as follows: mass range: 400–2500 *m/z*; AGC mode: fixed; enhanced Fourier transform: off; Emeter averaging: 0; microscans: 1.

MS² data acquisition was performed in ensemble mode for <17 kDa proteoforms. Prior to data acquisition, the AutoPiMS algorithm generates a list of spatial bins with assigned target *m/z* isolation windows corresponding to proteoform targets. The spatial locations of the bins in the list are then converted into chronological events in a MS² acquisition sequence calculated by the probe scan rate (typically maintained at the same rate compared to the survey). The list is then imported into an instrument method in XCalibur ("Targeted MS²" on Orbitrap Exploris 480) for MS² data acquisition.

MS² line scans in the automated MS² experiments were performed by shifting 20 μm longitudinally from the survey line scans (Fig. 1a) at same probe scan rate (2–4 μm/s). The close spacing between the lines guarantees the similarity of the spatial protein abundances between the survey and the MS² sampling. We note that MS² experiments can be performed in a spatially targeted manner without going through the entire line region (*e.g.*, Fig. 1e). During the MS² line scan, targeted MS² experiments were performed using HCD at a MS resolution of 30,000 at *m/z* 200 corresponding to an MS acquisition rate of 14.5 spectra/s (HCD pressure setting = 1)[17]. Typical HCD normalized CE and maximum injection time used in this study were 25–50 eV and 40 ms, respectively. Additional parameters for MS² in ensemble mode were the same as that for I²MS mode except for enhanced Fourier Transform enable: On. The ensemble MS² data for <17 kDa proteoform targets was kept in the.RAW file format for subsequent database searching.

**MS/MS of >17 kDa proteoforms on tissue.** Data acquisition for fragmentation-based analysis of >17 kDa proteoforms was performed on a Q-Exactive Plus mass spectrometer (Thermo Fisher Scientific, Bremen, Germany) with a modified central electrode voltage of 1 kV[36]. These experiments may be performed on the Exploris 480 system as well. The MS¹ survey line scan in I²MS mode was performed using identical parameters as the PiMS imaging experiments described in the "*PiMS imaging data acquisition and processing*" section at a probe scan rate of 2–4 μm/s. In particular, MS injection time was set 0.4 ms, and the MS resolution was set at 70,000 at *m/z* 200 corresponding to 1 s Orbitrap detection period. The HCD pressure setting was kept at 0.5 (arb., UHV pressure <5 × 10⁻¹¹ Torr). Additional relevant data acquisition parameters were as follows: mass range: 400–2500 *m/z*; AGC mode: fixed; enhanced Fourier transform: off; Emeter averaging: 0; microscans: 1.

Data acquisition for fragmentation-based analysis of >17 kDa proteoforms was performed following a similar logic and workflow as <17 kDa proteoforms as described in the above section ("*MS/MS of <*

17 kDa proteoforms on tissue"). However, for >17 kDa proteoform targets, MS² data was collected in I²MS mode processing for fragment ion charge assignment and mass-domain spectrum construction as described previously[36]. The list of target *m/z* isolation windows and the corresponding chronological events is then imported into an instrument method in XCalibur ("DIA" on Q-Exactive Plus) for MS² data acquisition.

MS² experiments in I²MS mode for >17 kDa proteoform targets using automated fragmentation were performed at MS resolution of 70,000 at *m/z* 200 corresponding to an Orbitrap detection period of 1 s (HCD pressure setting = 0.5)[17]. HCD collision energy (CE) and injection time were optimized to obtain fragments in the individual ion regime. Typical CE and injection time used in this study were 9–12 eV and 500–700 ms, respectively.

Post MS² data acquisition, an automated data processing pipeline was employed to find and sum all.stori files from the dataset corresponding to the same target for separate I²MS processing to generate mass-domain spectra. For each proteoform target, MS² data was first subjected to I²MS processing for fragment ion charge assignment and mass-domain spectrum construction as described previously[36]. All processed MS² spectra were stored in the.mzML format for subsequent database searching.

### Database search for AutoPiMS

ProSight Native (Proteinaceous, Evanston, IL) was used to search target proteoforms with the "batch mode" option in the "Native Proteoform" workflow. A UniProt.xml file containing all known Swiss-Prot accession numbers in the human protein database was input as the protein database (https://www.uniprot.org/proteomes/UP000005640). An annotated proteoform search was performed with a 100 Da precursor mass tolerance using a 10 ppm fragment mass tolerance. All proteoform IDs were manually validated and confirmed.

Starting from MS² data import, the database search workflow deviates for the ensemble MS² data type for <17 kDa and >17 kDa proteoform targets. In particular, for MS² of <17 kDa proteoforms (ensemble datatype), an automated data import function was implemented in ProSight Native to find and sum all the MS scans in the.RAW file corresponding to the same target precursor mass for searching. To curate the database search results, we implemented a FDR control step according to the B-H procedure[7] using *Eq. 2*, where *i* = ranking of the *p*-score, *Q* = user-defined FDR, *m* = total number of proteoforms in the database. We used *Q* = 1%, *m* = 1 M to calculate the B-H critical value for *p*-score of each proteoform ID, and find the proteoform IDs with B-H critical value > *p*-score. These proteoform IDs were considered as passing the 1% FDR control. Additional proteoform IDs were manually validated using TDValidator (Proteinaceous, Evanston, IL). Subsequence search with a 10 ppm mass tolerance was supplemented to find candidate truncated proteoforms. All truncated proteoforms found in the "Subsequence Search" mode were manually validated using TDValidator. A minimum of three matching fragment ions were required for a candidate proteoform to be returned as a hit. Proteoform IDs that did not pass the 1% FDR control and manual validation were listed as "non-FDR-controlled candidate proteoforms" in Supplementary Tables 6–8 within Supplementary Data 1. TD Validator options for fragment annotation are listed as follows: Validator type: "Simple"; Max PPM Tolerance: 5; Sub PPM Tolerance: 3; Minimum Score: 0.5.

For MS² of >17 kDa proteoforms (I²MS datatype), a modified THRASH algorithm[38] was applied to the spectra to determine fragment ion masses with a signal-to-noise cutoff of three. For each proteoform target, the database search generates a list of candidate IDs, and the best matched ID was manually selected by comparing the quality of the MS² spectra of the candidates, which serves as an FDR control step. Proteoform IDs were reported in the form of "number of detected fragments". TD Validator options for I²MS-type fragment annotation

are listed as follows: Validator type: "Shape"; Max PPM Tolerance: 5; Sub PPM Tolerance: 3; Minimum Score: 0.25. As an additional step to capture more detected fragments from the same MS² data, for each curated proteoform ID, the corresponding MS² scans in the.RAW file were averaged as a composite spectrum from which "ensemble-like" fragments were annotated. The number of additional fragments captured using this approach is listed as ".RAW average" in Supplementary Tables 3–5 within Supplementary Data 1, where the number of fragments solely detected in I²MS datatype is listed as "I²MS". The number of shared fragments from two approaches is listed as "shared".

### Intact Mass Tag (IMT) Search & Gene Ontology (GO) analysis

Mass-domain spectra from the LFQ experiment were converted to.mzML format and processed using a custom version of TDValidator (Proteinaceous, Evanston, IL) implemented with an MS¹ IMT search function as described in previous publications[16]. The spectrum was systematically calibrated according to the accurate masses of six MS/MS identified proteoforms in the 4–50 kDa mass range. The spectrum was searched against an ovarian cancer database, which was generated by the National Cancer Institute Clinical Proteomic Tumor Analysis Consortium (CPTAC) containing 9239 proteins (CPTAC, PDC study ID: PDC000113 & PDC000114, URL)[39]. Methionine on/off, water loss, monoacetylation, monophosphorylation and their one-to-one combinations were considered as possible proteoform modifications in the database. IMT search was performed with a ± 1.5 ppm mass tolerance.

GO analysis was performed using Metascape (https://metascape.org/)[40]. Specifically, a list of Entrez Gene IDs was retrieved for the 73 MS² identified proteoforms from Uniprot and submitted to Metascape for GO analysis. The result contains the top-level enriched GO biological pathways and Protein-protein interaction networks.

### Statistics & reproducibility

No statistical method was used to predetermine sample size. No data were excluded from the analyses. The experiments were randomized. The investigators were blinded to allocation during experiments and outcome assessment.

### Reporting summary

Further information on research design is available in the Nature Portfolio Reporting Summary linked to this article.

## Data availability

All data generated in this study (.raw files of the AutoPiMS, imaging, and LFQ experiments) have been deposited and are available on the MassIVE repository (https://massive.ucsd.edu/) with the identifier MSV000092418 [https://doi.org/10.25345/C5TM72B3K]. Source data of Fig. 1c–e and Fig. 2b are provided as a Source Data file. Source data are provided with this paper.

## Code availability

STORIBoard, which can process I²MS data and create output spectra, is a free program available from Proteinaceous. Custom compiled code used to process and create I²MS files for this study is available as Supplementary Data 3, and additional software and data that support the findings of this study are available from the corresponding author (N.L.K.), and requests will be fulfilled within 4 weeks.

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

## Acknowledgements

This study was supported by NIH P41 GM108569 (N.L.K.), P30 DA018310 (N.L.K.), P30 CA060553 (awarded to the Robert H. Lurie Comprehensive Cancer Center), F31 AG069456 (JPM), U.S. Department of Defense—Uniformed Services University of the Health Sciences (HU0001-16-2-0006, HU0001-19-2-0031, HU0001-20-2-0033, and HU0001-21-2-0027 (N.W.B., G.L.M., T.P.C.)) and NIH HuBMAP grant UH3CA246635 (N.L.K.). The views expressed herein are those of the authors and do not reflect the official policy of the Uniformed Services University of the Health Sciences, the Henry M. Jackson Foundation for the Advancement of Military Medicine, Inc., the Department of Army/Navy/Air Force, Department of Defense, or U.S. Government. Mention of trade names, commercial products, or organizations does not imply endorsement by the U.S. Government.

## Author contributions

J.P.M., P.S., J.O.K., N.L.K. conceptualized the study. J.P.M., P.S., K.R.D., M.A.R.H., J.O.K., N.L.K. developed the methodology. J.P.M., P.S., K.R.D., M.A.R.H. developed the algorithms, code, and software. P.S. performed the experiments. J.P.M. and P.S. analyzed the data and prepared the figures. T.P.C., N.W.B., G.L.M., R.T.F., R.D.M., J.M.C. contributed to the resources. J.O.K. and N.L.K. supervised the project. J.P.M., P.S., J.O.K., and N.L.K. contributed to the writing of the original draft. All authors contributed to the review and editing of the draft.

## Competing interests

N.L.K., K.R.D, and J.O.K. report a conflict of interest with individual ion technology and the development of software for processing resulting data. T.P.C. is a Thermo Fisher Scientific Inc. SAB member and receives research funding from AbbVie, Inc. The remaining authors declare no competing interests.
