## [Peer Review File · Nature Communications]

REVIEWERS' COMMENTS

Reviewer #1 (Remarks to the Author)

Manuscript by McGee et al. describes an exciting and novel approach to characterize proteoforms in localized tissue regions addressing an important but understudied area of spatial biology. The authors present an expansion of their recent workflow for proteoform informed imaging (PiMS). Previously, a Q Exactive Plus with single ion (I2MS) detection schemes was employed for mass spectrometry imaging (MSI), coined PiMS, and ion images of proteoforms up to ~70 kDa were acquired via nanospray desorption electrospray ionization (nanoDESI) MSI at 100 μm by 150 μm resolution (<https://www.science.org/doi/10.1126/sciadv.abp9929>). This permitted annotation of 42% of the 400 proteoforms detected by nanoDESI via external human kidney experimental databases and conformation of over a dozen proteoforms with in-situ fragmentation by HCD. Herein, the authors boosted the spatial resolution of PiMS, this time on an Exploris 480, to 20 μm (i.e., near cellular) resolution within the line scan and 80 μm lateral step size. They use several different instruments to complete one of the three workflows: PiMS (i.e., nanoDESI-MSI), label-free quantitation (LFQ), or AutoPiMS workflow. The latter is a novel and ambitious approach for high-quality spatially resolved MS_n suitable for continuous flow ambient ionization probes such as nanoDESI, DESI, etc. that has a potential to reshape in-situ MS_n for MSI applications if automated and made broadly applicable. Current implementation is however semi-automated, as it takes nanoDESI-MS1 outcome as input to generate MS method to acquire nanoDESI-MS2. Regardless, it represents an important step towards spatially resolved proteomics, and more specifically proteoform imaging, which is an essential tool in spatial biology toolkit.

The approach is technically sound and uses best-practice methods yielding high-quality data. However, the writing lacks clarity, material presented is quite complex and difficult to follow. The abstract would suggest high-resolution images of 1000 proteoforms were produced, which is misleading. The overall workflow can be broken into three major areas: (1) traditional MSI with single acquisition per pixel at ~20 μm by 80 μm resolution, (2) spatial label free quantitation (LFQ) with ~50 acquisitions per pixel at 80 μm by 80 μm resolution, and (3) AutoPiMS, which completes a MS1 survey scan (either in normal mode or I2MS mode) followed by MS2 (either in real-time or after some level of post-processing, this is unclear). The authors should rewrite the abstract and all relevant sections for clarity and explain how PiMS, LFQ, and AutoPiMS workflows intersect and interact. Including a schematic early on would be effective as it would help frame the reader's expectations and clarify where newly introduced AutoPiMS is used and how it is implemented. Below are several comments directed to each of the three major areas of this report.

1) Mass spectrometry imaging, MSI:

The PiMS (i.e., MSI experiment) was accomplished on an Exploris 480 (high-field Orbitrap) with 512 ms transient acquisitions. The authors do not make it explicitly clear if I2MS detection is enabled on the Exploris 480, but one would assume yes based on the recent report and PiMS acronym being somewhat synonymous to I2MS-MSI due to previous implementation on the Q Exactive Plus (low-field Orbitrap). The methods are hard to follow with 3 Orbitraps being used for multiple workflows. The authors should consider breaking methods into 3 different sections (corresponding to 3 different workflows) for clarity.

400 proteoforms were detected on the Q Exactive Plus in earlier report. How many were detected on the Exploris? Given greater ion flux, one would expect higher sensitivity, which should translate into greater depth of coverage. If I2MS was employed (and it is unclear if it was), the benefit should have been even greater.

The LFQ experiment, which is not an MSI experiment, represents a major outcome described in this report (with >1000 proteoforms). However, a less-than-careful reader might easily come away believing these are all MSI results and MSI produced 1000 images in the traditional PiMS experiment, which is clearly not the case. In reality images were obtained for much smaller number of proteoforms. Similarly, the first impression is that the Figure 1 line scan workflow was done for everything, hence the name "AutoPiMS", but apparently this is not the case. The inclusion of the MSI data is also confusing. Given the focus on LFQ of the 1013 features, why choose just these 17 images? How many features were annotated in AutoPiMS that were detected in PiMS, and furthermore, how many were detected in AutoPiMS survey scans that were not detected with PiMS?

Any mention of near cellular resolution has the caveat that the width of the pixel is 80 μm , which can easily contain several if not over a dozen cells. Any mention of resolution should clearly state

that it is not a square pixel, but it is $\sim 20 \mu\text{m}$ by $80 \mu\text{m}$ pixel, and even then, it is variable in nanoDESI experiment.

2) Spatial label free quantitation (LFQ):

Averaging ~ 50 acquisitions per "pixel" for LFQ is interesting terminology as for this to be a "pixel" there should be images generated from this analysis. While the broader area MSI was included, no images were presented from the LFQ workflow. This is a comparable level of sampling to laser capture microdissection (LCM) based approaches and images have been produced from LCM-LCMS, why not here? A recent nanoDESI report claiming cellular resolution annotated a couple dozen proteoforms (<https://pubs.acs.org/doi/abs/10.1021/acs.analchem.2c04795>). Given the large number of annotations in this report, there must be a comment on proteome coverage of MSI compared to LFQ. Were MSI annotations at 5-10% of LFQ annotations? 10-20%?

Methods state stages were moved at $3\text{-}5 \mu\text{m/s}$ for LFQ, but then specifically in LFQ section $4 \mu\text{m/s}$ with 2 spectra/s (512 ms transients on the Exploris) was stated. Given ~ 50 spectra per Figure 2, this means a maximum of 25 seconds of travel per $80 \mu\text{m} \times 80 \mu\text{m}$ "pixel". After forming the junction on the tissue, it would take 20 s for the probe to move $80 \mu\text{m}$, meaning only 40 acquisitions, or the actual sampled area was $80 \mu\text{m}$ by $100 \mu\text{m}$. Assuming the length of the liquid microjunction was the same as the reported width ($80 \mu\text{m}$) and given it will take 20 s to move the probe $80 \mu\text{m}$ to a new area, this would be an extreme amount of oversampling on the tissue. The authors should comment on this.

The authors state reproducibility for technical replicates. However, how do the authors account for the diminishing signal if a nanoDESI probe was allowed to sit on a surface for an extended period? Large number of acquisitions with blank signal could artificially inflate high abundance proteoforms and reduce low abundance proteoforms.

3) AutoPiMS:

PiMS was defined as "Proteoform imaging Mass Spectrometry", and "AutoPiMS" has a strong connotation that is directly in-tandem with PiMS (i.e., MSI) but no more than one line of AutoPiMS was demonstrated. How automated is this technique? It appears manual inclusion of the lists in Xcalibur is required. ("The list is then imported into an instrument method in XCalibur...for MS2 data acquisition.") It is unclear if one line scan needs to be done, or the entire MSI needs to be done to implement AutoPiMS? Given the amount of oversampling this would also impact subsequent "survey scans" or implementation in MSI line scans. Can the authors comment on how to take this from proof-of-concept to actual implementation for more than one line scan? AutoPiMS was completed at roughly half the speed of the PiMS. How long would the theoretical AutoPiMS of every line scan on the Exploris take? Information on all the timings should be included to allow for some practical estimates on the overall experimental time. Using the parameters provided, simple calculation suggests >10 hrs. How long can the tissue be at ambient temperature before there are noticeable changes due to oxidation or truncations or cleavages from proteases? From the Figure 2, it is unclear if AutoPiMS was performed on the same area of tissue that had LFQ completed.

Specific comments:

Abstract: "automated molecular histology approach" is cryptic and uninformative. This statement suggests the approach yielded images of 1000 proteoforms, which is misleading. Rephrase and clearly explain the intricacies of the approach resulting in different depth of coverage associated with different workflows.

Lines 38-42: DDA MALDI-MSn was previously demonstrated for lipids (<https://www.nature.com/articles/s41592-018-0010-6>) and could theoretically be accomplished for proteoforms in a similar fashion. Some more recent approaches (e.g., PASEF) have made huge strides for in-situ MSn.

Lines 42-43: Reads fragmented and does not introduce nanoDESI as written.

Line 50: This is "nanoDESI" and not "DESI" (and these common abbreviations should be introduced). Or did the authors mean to reference FAIMS-DESI-MSI report where CID and UVPD were used to confirm several proteoforms up to ~ 20 kDa (<https://pubs.acs.org/doi/10.1021/acs.analchem.8b00967>)?

Lines 51-55: This paragraph introduces I2MS, then launches directly into the AutoPiMS and LFQ, leaving the reader wondering as how AutoPiMS fits into the MSI workflow and connects with LFQ. What does "engine for molecular identification" refer to? The reader would benefit from breaking this down and clearly explaining the workflows employed and their connections and interdependencies.

Lines 63-70: Why are scan rates for the stages and MS2 speed stated here and then again in

methods section? This led to some inconsistencies and should be avoided. There are several different nanoDESI analyses included in this report, and AutoPiMS is only a part of the story. There should be a much clearer description of how these separate workflows play and integrate. (Also, no need to introduce abbreviation for Hertz.)

Lines 101-109: 87 of 113 proteoforms, yet favorable windows only for 25? Are these 25 a subset of larger proteoforms within the dataset? Supplementary figure 4 suggests these proteoforms were detected with incredibly low SNR (MS1), which is concerning. Please comment.

Line 110 (and elsewhere): Replace MS2 spectra with tandem MS or fragment spectra

Line 118: Supplementary figure 6 seems to be out of context. It is unclear how this connects to regular MS2 and I2MS-MS2. What is the take home message?

Line 119: 1% relative abundance of a noisy spectrum is not informative. SNR might be better performance metrics than relative abundance.

Lines 117-127: Clarify which instruments are being used. One could assume Q Exactive for the "ensemble mode" but was it the Plus or the HF? Provide some explanation as why so many different instruments have been used as this could potentially limit the applicability. What is meant by technical replicates in this context? Also, comment on reproducibility among replicates.

Line 152: Was AutoPiMS employed for LFQ? If so, how exactly was this accomplished? This information is not directly stated anywhere other than Figure 2a where it is unclear. This is important because of potential oversampling effects on downstream quantitation.

Figures are overly complicated and therefore difficult to follow and connect to explanations and conclusions provided in the text. Some of this material can be moved to SI to make the overall flow of the experiment clearer and connect better to the text. For example, Figures 1c and 1d could be moved to SI without losing any context and same goes for graphical fragment maps. In this case, less is more. In Figure 2a, was AutoPiMS applied only along the vertical dashed line (and not implemented for the PiMS region)? How can one be certain that a monomethylation is present solely on Arg68 if only one AutoPiMS vertical line scan was applied? Terminology is also confusing, "pixels" would suggest that the LFQ experiment could create images. If so, why not include these images? It is challenging to gauge small differences in the images with Jet or other color schemes, consider changing to viridis or ciridis. Overlaying of Jet with a red-scale image makes figure 2f hard to read.

Figure 2 and related text: Ideally, the report should have included a validation of selected proteins (and/or PTMs) using complementary techniques.

Supplementary Information:

Just as the main body of the manuscript, the methods were hard to digest with three instruments with different settings used for different purposes. It is challenging to find the information with current organization (for example, identify parameters specific to the Q Exactive HF). One way to improve readability is to group all MS settings and relevant acquisition parameters into their respective "MSI", "LFQ", "AutoPiMS" section.

All supplementary tables are of high-quality, but it is unclear why is SNR reported only in selected tables.

Supplementary Figure 2: Challenging to judge what 1% relative intensity corresponds to. Perhaps zooming on the lowest abundance would be enough to inform the reader? Specify which instrument was this acquired on, and which workflow was used: the LFQ on Exploris, MSI on Exploris or AutoPiMS demonstration survey scan on Q Exactive Plus (I2MS or normal). Reader shouldn't have to try to retrieve this information going back and forth between SI and the main body of the manuscript.

Supplementary Figure 3: Same as above, which experiments do these spectra belong to?

Supplementary Figure 4: It is challenging to see any of the intensities in (b), and the plot is quite hard to see given the red coloring.

Supplementary Figure 5: What is the SNR for these species? Does not seem to be above 5, and at 17% RA in (a), it does not seem to even be a peak? This is mildly concerning since it appears the algorithm isn't picking the peaks.

Supplementary Figure 6: Which 79-targeted proteoforms are captured here and where do they come from, LFQ or MSI?

Reviewer #2 (Remarks to the Author)

This study developed a novel technique, AutoPiMS, for the detection of spatial proteoforms in human tissues. This technique is impressive due to its ability to combine potentially several methods into one, enabling the characterization of proteoforms in a 4-dimensional manner. Importantly the technique provides an opportunity to delineate (with accuracy and speed) proteoforms associated with tumour vs stroma cells with potential applications in many other solid tumours especially those in which the interplay between tumour and stroma cells have significance in therapy and patient outcome. The article is well written, detailed and with novel findings and as such should be published.

One minor comment is for the authors to consider briefly highlighting the (potential) clinical significance of the differential expression of CRIP1 in tumour regions. Some studies (such as <https://www.ncbi.nlm.nih.gov/pmc/articles/PMC8820892/>) have alluded to this.

Reviewer #3 (Remarks to the Author)

The significance of this study lies in the development of an integrated platform, AutoPiMS, that drives advancements in proteoform-level spatial biology. The platform enables four-dimensional characterization of proteoform signatures: intact molecular mass, spatial distribution, quantitative analysis of differential expression, and molecular identification. In particular, the automated data acquisition engine enables proteoform identification up to ~54 kDa at a speed of <1 minute per proteoform directly off tissue. A spatially resolved study of ovarian cancer tissue with 20-micron resolution identified >300 differential proteoforms in stroma versus tumor regions from the same patient, including differential localization of methylated forms of CRIP1. This platform fills the gap between high-confidence proteoform discovery and spatial proteomics, opening up a new avenue for discoveries and precision diagnostics in clinical histology.

I have a minor question.

Have the researchers compared their findings to existing methods or performed any validation?

It would be nice to have a specific comparison of the advantages and disadvantages of the autoPiMS method compared to traditional immunohistochemistry results.

RESPONSE TO REVIEWERS' COMMENTS

Reviewer 1

Excerpted General Comments:

Manuscript by McGee et al. describes an exciting and novel approach to characterize proteoforms in localized tissue regions addressing an important but understudied area of spatial biology. The authors present an expansion of their recent workflow for proteoform informed imaging (PiMS). Previously, a Q Exactive Plus with single ion (I2MS) detection schemes was employed for mass spectrometry imaging (MSI), coined PiMS, and ion images of proteoforms up to ~70 kDa were acquired via nanospray desorption electrospray ionization (nanoDESI) MSI at 100 μm by 150 μm resolution (<https://www.science.org/doi/10.1126/sciadv.abp9929>). This permitted annotation of 42% of the 400 proteoforms detected by nanoDESI via external human kidney experimental databases and conformation of over a dozen proteoforms with in-situ fragmentation by HCD. Herein, the authors boosted the spatial resolution of PiMS, this time on an Exploris 480, to 20 μm (i.e., near cellular) resolution within the line scan and 80 μm lateral step size. They use several different instruments to complete one of the three workflows: PiMS (i.e., nanoDESI-MSI), label-free quantitation (LFQ), or AutoPiMS workflow. The latter is a novel and ambitious approach for high-quality spatially resolved MSn suitable for continuous flow ambient ionization probes such as nanoDESI, DESI, etc. that has a potential to reshape in-situ MSn for MSI applications if automated and made broadly applicable. Current implementation is however semi-automated, as it takes nanoDESI-MSI outcome as input to generate MS method to acquire nanoDESI-MS2. Regardless, it represents an important step towards spatially resolved proteomics, and more specifically proteoform imaging, which is an essential tool in spatial biology toolkit.

Response: We heartfully thank the reviewer for these positive comments, and for their time and care in evaluation of the work. We respond below with the general aspiration to match the energy with detailed, earnest revisions.

Detailed Comments:

1) The approach is technically sound and uses best-practice methods yielding high-quality data. However, the writing lacks clarity, material presented is quite complex and difficult to follow.

Response: We thank the Reviewer for the constructive feedback for the writing structure of the paper. As the Reviewer will find in this revision, the manuscript structure and the materials presented has been carefully thought through and revised to best reflect clarity according to the reviewer's suggestions. In particular, we have rewritten the **Abstract** (147 words), expanded the introduction paragraphs to introduce AutoPiMS as a novel spatially-resolved on-tissue proteoform identification workflow (Line 50-69), reformulated the walkthrough paragraph of the AutoPiMS workflow (Line 134-147), clarified the relationship among datatypes presented for the AutoPiMS-enabled spatial proteoform biology study (Line 192-240). The Reviewer will find detailed changes in the track-change version of the manuscript and the Responses to specific questions in this letter in the following.

2) The abstract would suggest high-resolution images of 1000 proteoforms were produced, which is misleading.

Response: We indeed detected circa 1000 proteoforms by label-free quantitation from the human ovarian cancer tissues studied by PiMS. We have revised the **Abstract** to clearly define the differences among proteoforms 'detected', 'identified' and found to be 'differentially expressed' (by label-free quantitation), what we believe to be the source of the issues and comments about the putatively misleading passages. Here is what we have put forward in the revised **Abstract** and aligned the manuscript content with this change (Line 34): "*From a total of ~1000 proteoforms detected by region-of-interest label-free quantitation, we discovered 303 differential proteoforms in stroma versus tumor from the same patient.*"

3) The overall workflow can be broken into three major areas: (1) traditional MSI with single acquisition per pixel at ~20 μm by 80 μm resolution, (2) spatial label free quantitation (LFQ)

with ~50 acquisitions per pixel at 80 μm by 80 μm resolution, and (3) AutoPiMS, which completes a MSI survey scan (either in normal mode or I2MS mode) followed by MS2 (either in real-time or after some level of post-processing, this is unclear). The authors should rewrite the abstract and all relevant sections for clarity and explain how PiMS, LFQ, and AutoPiMS workflows intersect and interact. Including a schematic early on would be effective as it would help frame the reader's expectations and clarify where newly introduced AutoPiMS is used and how it is implemented. Below are several comments directed to each of the three major areas of this report.

Response: We would like to make the clarification that AutoPiMS refers to the newly-developed workflow for on-tissue multiplexed MS² to identify and characterize intact proteoforms. These proteoforms were discovered from multifaceted PiMS-derived datatypes including full MS¹ profile, proteoform images, and label-free quantitation (LFQ). As the reviewer pointed out in their summary, we have revised the manuscript systematically to make these clarifications. In particular, we have rewritten the **Abstract** to clarify the focus on AutoPiMS as an on-tissue multiplexed MS² workflow. Moreover, we kept the focus of **Fig. 1** and the first half of the manuscript (before Line 151) on the technical development of AutoPiMS workflow, and pivot into spatial biology study (**Fig. 2**) using multifaceted PiMS-derived techniques including imaging, LFQ, and AutoPiMS. Detailed changes to the manuscript are listed below in point-by-point responses to Reviewer's comments.

3)-1 Mass spectrometry imaging:

MSI: The PiMS (i.e., MSI experiment) was accomplished on an Exploris 480 (high-field Orbitrap) with 512 ms transient acquisitions. The authors do not make it explicitly clear if P²MS detection is enabled on the Exploris 480, but one would assume yes based on the recent report and PiMS acronym being somewhat synonymous to P²MS-MSI due to previous implementation on the Q Exactive Plus (low-field Orbitrap). The methods are hard to follow with 3 Orbitraps being used for multiple workflows. The authors should consider breaking methods into 3 different sections (corresponding to 3 different workflows) for clarity.

Response: We have added a sentence in the **Methods** section to clarify that I²MS detection was enabled on the Exploris system: “*The Orbitrap mass analyzer on Exploris 480 system operates at a central electrode voltage of 4 kV, allowing for more favorable ion lifetimes for I²MS over models that operate at 5 kV.*” To offset the complexities from the use of multiple instruments, we have reorganized and rewritten the **Methods** section to improve the clarity for each workflow discussed in the manuscript. Now, the revised **Methods** section is composed of four main sections: PiMS imaging (“*PiMS ion source and sampling conditions*” and “*PiMS imaging data acquisition and processing*”), label-free quantitation (“*Label-free quantitation (LFQ)*”), AutoPiMS (“*AutoPiMS algorithms*”, “*AutoPiMS data acquisition*”, and “*Database search for AutoPiMS*”), and “*Intact Mass Tag (IMT) Search & Gene Ontology (GO) analysis*”. In the AutoPiMS section, we start from introducing the algorithms of how the MS/MS methods are generated followed by detailed MS/MS data acquisition for direct infusion mixture, <17 kDa proteoforms on tissue, and >17 kDa proteoforms directly from tissue.

400 proteoforms were detected on the Q Exactive Plus in earlier report. How many were detected on the Exploris? Given greater ion flux, one would expect higher sensitivity, which should translate into greater depth of coverage. If I²MS was employed (and it is unclear if it was), the benefit should have been even greater.

Response: In this imaging dataset, we detected 618 proteoforms above 0.1% relative abundance (compared to ~400 observed in the earlier report from Su et al., *Sci. Adv.*, 2022). To address the reviewer’s question, we revised the manuscript (Line 200) and added a full spectrum and a list of 618 proteoform mass values detected in this imaging dataset: “*In this imaging dataset, 618 proteoforms were detected and imaged above 0.1% relative abundance (Supplementary Table 14 and Supplementary Fig. 10).*” We have also clarified in the **Methods** section that I²MS was employed for this imaging experiment as mentioned in the response above.

3)-2 The LFQ experiment, which is not an MSI experiment, represents a major outcome described in this report (with >1000 proteoforms). However, a less-than-careful reader might easily come away believing these are all MSI results and MSI produced 1000 images in the

traditional PiMS experiment, which is clearly not the case. In reality images were obtained for much smaller number of proteoforms.

Response: As noted above, we agree with the Reviewer on the confusion between LFQ and imaging results. Given the multiple areas of advancement within this manuscript, we have moved to clarify the workflows and datasets emerging from them. So, in the revised manuscript, we report clearly that PiMS imaging of the different regions of interest on the same tissue section, from the same patient, resulted in the detection and imaging of a total of 618 proteoform masses (Line 199): *“In the next step, we compared the LFQ results with a PiMS imaging experiment performed on a region with spatially comingled tumor and stromal compositions (Fig. 2a). In this imaging dataset, 618 proteoforms were detected above 0.1% relative abundance (Supplementary Table 14 and Supplementary Fig. 10).”*

Similarly, the first impression is that the Figure 1 line scan workflow was done for everything, hence the name “AutoPiMS”, but apparently this is not the case. The inclusion of the MSI data is also confusing. Given the focus on LFQ of the 1013 features, why choose just these 17 images?

Response: We agree with the Reviewer and revised the manuscript to better explain the relationship between the samples and the panels in the figures. We used the first patient sample (~95% tumor) to demonstrate AutoPiMS as a multiplexed on-tissue MS/MS workflow to identify proteoforms, and all display items in **Fig. 1** were obtained from that first patient sample. In **Fig. 2**, we moved on to the second patient sample of higher complexity (comingled tumor and stroma regions). For this complex sample, we employed the set of PiMS-related workflows to understand its spatial biology, including imaging, LFQ to assert differential proteoform expression and MS/MS identification using AutoPiMS. AutoPiMS here was employed to specifically identify and characterize the 17 proteoform signatures with images displayed in **Fig. 2b** that are differentially detected in tumor and stromal regions. Despite 1013 proteoforms *detected* in LFQ, some of these proteoforms did not contain enough ion counts to generate an informative, quality image or did not show distinct features in their PiMS images. Moreover, the imaging experiment was performed on a different region of the same tissue compared to LFQ, which may contain a different set of proteoforms due to the spatial heterogeneity. We have

revised the first sentence describing **Fig. 2** to make the clarification as follows (Line 177): “We deployed AutoPiMS to **identify proteoform signatures** in histology-defined tumor versus stromal regions within HGSOC tissue from a single patient.”

Also these changes on Line 199 and Line 209 were also made, respectively: “In the next step, we compared the LFQ results with a PiMS imaging experiment performed on a region with spatially comingled tumor and stromal compositions (**Fig. 2a**). In this imaging dataset, 618 proteoforms were detected above 0.1% relative abundance (**Supplementary Table 14** and **Supplementary Fig. 10**)... **Fig. 2b** shows PiMS images of 17 of the proteoforms with highly differential ion counts between tumor and stromal regions in LFQ.”

(Line 209): “AutoPiMS was subsequently employed for direct top-down MS/MS of these proteoform signatures... All 17 proteoforms were detected in the survey scan, 16 of them were MS¹ annotated, and 14 of them were MS² identified (**Supplementary Table 15**).”

How many features were annotated in AutoPiMS that were detected in PiMS, and furthermore, how many were detected in AutoPiMS survey scans that were not detected with PiMS?

Response: We reasoned that the Reviewer was asking particularly about the results shown in **Fig. 2** as a follow up to the previous comment. As explained in the previous response, we used AutoPiMS to selectively target the 17 proteoforms with PiMS images shown in **Fig. 2b**. In this experiment, 14 of the 17 PiMS-detected proteoforms were identified by AutoPiMS. All proteoforms detected in AutoPiMS survey scan were detected in PiMS. To further clarify, we have also added a table of the information for the 17 proteoforms as **Supplementary Table 15** in the revised package. The particular part of the manuscript revised to address the Reviewer’s comments is as follows (Line 209): “AutoPiMS was subsequently employed for direct top-down MS² of these 17 signature proteoforms... All 17 proteoforms were detected in the survey scan, 16 of them were MS¹ annotated, and 14 of them were MS² identified (**Supplementary Table 15**).”

Any mention of near cellular resolution has the caveat that the width of the pixel is 80 μm , which can easily contain several if not over a dozen cells. Any mention of resolution should clearly state that it is not a square pixel, but it is $\sim 20 \mu\text{m}$ by $80 \mu\text{m}$ pixel, and even then, it is variable in nanoDESI experiment.

Response: We agree with the Reviewer that the line spacing of $80 \mu\text{m}$ should be specified in the manuscript when $20 \mu\text{m}$ lateral spatial resolution is mentioned. We have revised the manuscript accordingly (Line 236): *“Given the ~ 20 micron lateral spatial resolution and 80 micron line spacing, cell-specific observations as well as the functional role of Arg68me0 in angiogenesis will require future study.”*

3)-2 *Spatial label free quantitation (LFQ):*

Averaging ~ 50 acquisitions per “pixel” for LFQ is interesting terminology as for this to be a “pixel” there should be images generated from this analysis. While the broader area MSI was included, no images were presented from the LFQ workflow. This is a comparable level of sampling to laser capture microdissection (LCM) based approaches and images have been produced from LCM-LCMS, why not here?

Response: We agree with the Reviewer that “pixel” is a terminology indicating images are generated. LFQ experiments were performed in a similar way as the PiMS imaging experiments, which has been described in the revised manuscript and Methods section (“*Label-free quantitation (LFQ)*”), and will be explained in further responses to two Reviewer questions below. In contrast to the PiMS imaging experiments performed in complex tissue regions, LFQ experiments were designed to take hundreds of samples from well-defined tumor or stromal regions to construct the statistics for quantitative interpretations. Therefore, the 240 “pixels” in LFQ experiments highlighted in **Fig. 2a** did not intend to demonstrate the spatial differences in the pathological context (*e.g.*, tumor, stroma). Moreover, with the spatial resolution of up to $20 \mu\text{m}$, we do not assert any cell type or intratumor heterogeneities in this piece.

To conclude, we agree with the Reviewer that proteoform images can be generated from the LFQ dataset similar to laser capture microdissection coupled to LC-MS workflows, but we do not show LFQ-derived proteoform images in this particular workflow. To address the reviewer's concern about the terminology "pixel" used in the paper, we used "sampled region" instead in the revised manuscript for clarification. The Reviewer will find these changes in **Fig. 2a** and in Line 157, 183, and 187 in the **Fig. 2** captions and the manuscript. Moreover, we updated the LFQ depiction in **Fig. 2a** to better convey the experimental design.

A recent nanoDESI report claiming cellular resolution annotated a couple dozen proteoforms (<https://pubs.acs.org/doi/abs/10.1021/acs.analchem.2c04795>). Given the large number of annotations in this report, there must be a comment on proteome coverage of MSI compared to LFQ. Were MSI annotations at 5-10% of LFQ annotations? 10-20%?

Response: We appreciate the Reviewer for mentioning the recent advances in spatial proteoform annotation in the field. We ran a comparison between the MSI and LFQ annotated proteoforms using our algorithms. In particular, we overlaid the 618 MSI proteoforms with the 552 LFQ proteoforms passing 1% FDR and picked the ones within 3 Da of mass tolerance found that 249 of the MSI proteoforms were detected in LFQ, which corresponds to a 45% of proteoform overlap. The discrepancies in proteoform detection between MSI and LFQ is potentially due to the difference in linear dynamic range of the two approaches and the spatial heterogeneity of the sample. We have revised the paper accordingly on Line 202: "249 common proteoforms were found in both the imaging and the LFQ dataset, featuring a 45% overlap in proteome coverage."

Methods state stages were moved at 3-5 $\mu\text{m/s}$ for LFQ, but then specifically in LFQ section 4 $\mu\text{m/s}$ with 2 spectra/s (512 ms transients on the Exploris) was stated. Given ~50 spectra per Figure 2, this means a maximum of 25 seconds of travel per 80 μm x 80 μm "pixel". After forming the junction on the tissue, it would take 20 s for the probe to move 80 μm , meaning only 40 acquisitions, or the actual sampled area was 80 μm by 100 μm . Assuming the length of the liquid microjunction was the same as the reported width (80 μm) and given it will take 20 s to move the probe 80 μm to a new area, this would be an extreme amount of oversampling on the tissue. The authors should comment on this.

Response: We thank the Reviewer for thorough consideration and calculation for the LFQ experimental methodology. We would like to clarify that LFQ experiments was performed in exactly the same way as PiMS imaging experiment except the lower probe rastering rate ($4\ \mu\text{m/s}$ rather than $5\ \mu\text{m/s}$). During the experiment, the liquid bridge was moving continuously along the lateral direction to extract proteins while the mass spectrometer recorded a spectrum every 0.5 second. The dataset was reconstructed to generate data for each “sampled region” by binning adjacent ~ 50 spectra equivalent to $80\ \mu\text{m} \times 80\ \mu\text{m}$ areas. The experiment was **not** performed by reestablishing a new liquid bridge and reinitiating the liquid extraction process for an adjacent $80\ \mu\text{m} \times 80\ \mu\text{m}$ area and moving laterally for 20 seconds.

In LFQ experimental design, we intended to extract all proteoform signals from each sampled region without signal suppression caused by saturation. The figure below shows typical imaging profile at $5\ \mu\text{m/s}$ (top) and LFQ profile at $4\ \mu\text{m/s}$ (bottom). We did not observe signal saturation in the scans in imaging experiments, in which a higher protein concentration was analyzed (each unit volume of solvent extracts 20% more tissue area, and a 0.4 millisecond injection time was used rather than 0.2 millisecond in LFQ). Therefore, we believe that signal saturation rarely took place in LFQ experiments from these tissues.

Another important consideration for quantitation of each “sampled region” is to minimize the carryover of proteins from the previous “sampled region”. For LFQ experiments for this set of cancer tissues, we optimized the rastering rate of the probe to make sure that proteins are maximally extracted in every $80\ \mu\text{m} \times 80\ \mu\text{m}$ area with minimal carryover from the adjacent areas. In the figure shown above, we compared the temporal profile of total protein signal at a rastering rate of $5\ \mu\text{m/s}$ (imaging experiments) and $4\ \mu\text{m/s}$ (LFQ experiments). In contrast to even sampling profiles in imaging, LFQ profile shows many spike features with highest ion counts (at the tip of the spike) $\sim 10\%$ to 20% higher than the average profile in imaging experiment. These profiles correspond to an almost complete extraction of protein content within every $20\ \mu\text{m}$ of lateral distance. In this regard, when protein signals within $80\ \mu\text{m}$ distance are binned to construct a sampled region for LFQ quantitation, each region will contain minimal contribution from adjacent $80\ \mu\text{m}$ area. Another reason we chose $80\ \mu\text{m}$ as the lateral size of sample is based on the width of the liquid bridge ($80\ \mu\text{m}$), which eliminates the difference between lateral or horizontal orientation and mimic the process in LCM, micro-punch, and other microsampling approaches.

To address the reviewer’s concern, we have included the above figure and explanation in the **Methods** section and as **Supplementary Fig. 15**, and we have also revised the conceptual depiction in **Fig. 2a**.

The authors state reproducibility for technical replicates. However, how do the authors account for the diminishing signal if a nanoDESI probe was allowed to sit on a surface for an extended period? Large number of acquisitions with blank signal could artificially inflate high abundance proteoforms and reduce low abundance proteoforms.

Response: We appreciate the focus here, as this is the first report of using individual ion MS to perform LFQ. The nano-DESI probe was constantly moving during the sampling process and did not park on a surface for an extended period of time. During the entire sampling process for LFQ, the MS injection time was constant, and the rastering rate was set to extensively sample proteins from each $80\ \mu\text{m} \times 80\ \mu\text{m}$ sampled region in the tissue section. We reasoned that the reviewer was referring to the situation where the probe was allowed to park on the tissue for long

time, and the initial MS scans would be saturated and dominated by high abundance proteoform signals. However, our experiments were performed at a rastering rate slightly lower than the imaging experiment, in which less protein signals were sampled per scan compared to imaging. Therefore, we do not anticipate artificial inflation of high abundance proteoforms due to the probe operation mode, and indeed, we observed similar spectral dynamic range in LFQ experiment compared to imaging as evidenced by the number of proteoforms detected in each experiment.

3)-3 *AutoPiMS*:

PiMS was defined as “Proteoform imaging Mass Spectrometry”, and “AutoPiMS” has a strong connotation that is directly in-tandem with PiMS (i.e., MSI) but no more than one line of AutoPiMS was demonstrated. It is unclear if one line scan needs to be done, or the entire MSI needs to be done to implement AutoPiMS?

Response: We agree with the Reviewer that AutoPiMS workflow is directly in-tandem with PiMS. However, we would like to clarify that PiMS is a term that describes the scalable technique that enables spatial profiling of proteoforms in tissue. PiMS data can be obtained for a single line or for a region containing multiple lines. In a similar sense, AutoPiMS can be performed with PiMS experiments of different scales including a single line or a region for targeted proteoform identification.

Given the amount of oversampling this would also impact subsequent “survey scans” or implementation in MSI line scans. Can the authors comment on how to take this from proof-of-concept to actual implementation for more than one line scan?

Response: We agree with the Reviewer that oversampling is frequently observed during the survey line scan data acquisition. To obtain enough protein signal for MS² acquisition, we shifted the subsequent line scan by 20 μm as described in the main text (Line 76: “MS² fragmentation is then performed by running the PiMS probe across a fresh line parallel to the survey line scan but offset by ~20 μm”). This allowed us to obtain precursor proteoform signals by accessing a 20 μm-wide fresh tissue region with highly similar proteoform spatial distributions compared to the

adjacent survey line. In particular cases when precursor proteoform signal does not meet the abundance requirement for MS² experiments, we can further shift the probe by 50-75 μm. In the case of a survey imaging experiment with multiple parallel line scans, we reserve the space between the lines. MS² experiment will be performed in the staggered region containing fresh tissue area not scanned by the survey as illustrated in the figure below:

We have revised the manuscript to further explain these concepts (Line 64): “*AutoPiMS augments PiMS with a computational engine for unattended proteoform target selection and acquisition method generation, and is directly interfaced with high-throughput data processing and database search. AutoPiMS streamlines multiplexed on-tissue top down proteomics and can be readily interfaced with a variety of electrospray-based protein MS imaging modalities to extend proteome coverage in spatial proteomics, advancing the field of molecular histology.*”

and Line 82: “*We note that the first step in the AutoPiMS workflow is not limited to a single line scan but can also be applied to a PiMS imaging experiment with space between lines reserved for MS² data acquisition.*”

How automated is this technique? It appears manual inclusion of the lists in Xcalibur is required. (“The list is then imported into an instrument method in XCalibur...for MS² data acquisition.”)

Response: The data transfer steps and inclusion of the target list in XCalibur are not automated as the reviewer pointed out. All data acquisition, computational steps for target list generation and database searching are fully automated. We have revised the manuscript to clarify these points (Line 70): “*AutoPiMS achieves unattended identification of proteoforms using a semi-automated spatially-aware, data-dependent acquisition strategy.*”

and Line 81: “All steps aside from data transfer and MS² method setup are fully automated and can be customized manually as desired.”

AutoPiMS was completed at roughly half the speed of the PiMS. How long would the theoretical AutoPiMS of every line scan on the Exploris take?

Response: We performed AutoPiMS survey scan at 2-4 $\mu\text{m/s}$, and the line scan took 35-50 minutes to complete. In the line scan described in **Supplementary Fig. 6**, the survey line scan took 40 minutes to cover a 9.6 mm line region at 4 $\mu\text{m/s}$ rastering rate. The corresponding MS² experiment took identical time as the survey scan. Considering the computational steps (*e.g.*, MS² target list curation, datafile handling and transferring, database searching) and manual operation to shift the sample between the survey and the MS² lines, an entire AutoPiMS experiment takes 2-3 hours to finish. We have added this information in Methods section (section: “General schematic AutoPiMS workflow”) to address the comment. Moreover, we have also added a table in “PiMS ion source and sampling conditions” section to clarify the probe rastering rates we used for different experimental workflows:

Experiment type	Probe rastering rate ($\mu\text{m/s}$)
PiMS	5
Label-free quantitation (LFQ)	4
AutoPiMS	2-4

Information on all the timings should be included to allow for some practical estimates on the overall experimental time. Using the parameters provided, simple calculation suggests >10 hrs. How long can the tissue be at ambient temperature before there are noticeable changes due to oxidation or truncations or cleavages from proteases?

Response: We have compared the level of oxidation at 0 and 24 hours acquired on the same tissue section using Glyceraldehyde-3-phosphate dehydrogenase (GAPDH) as a reporter. As

shown in the figure below, canonical GAPDH proteoform (35.92 kDa, left) is oxidized at C152 by ambient oxygen evidenced by a mass shift of 32 Da corresponding to the addition of two oxygen atoms (35.95 kDa, right). The oxidated GAPDH was at 10% level of the canonical GAPDH after 24 hours of ambient air exposure. This proves that within the time frame of one AutoPiMS experiment (2-3 hours), the effect of tissue oxidation on the data quality was negligible.

Regarding protein truncation by proteases, we have applied successive ethanol washes to the tissue section prior to PiMS and AutoPiMS experiments, which fixes and precipitates the proteins (described in detail in **Methods** section: “*Tissue/sample preparation*”). With the ethanol wash, protease activities should be deactivated.

From the Figure 2, it is unclear if AutoPiMS was performed on the same area of tissue that had LFQ completed.

Response: AutoPiMS was performed along the dashed line region in **Figure 2**. We have revised the manuscript to clarify this (Line 210): “A survey line scan followed by an adjacent MS² scan were performed along the dashed line in **Fig. 2a** next to the LFQ-profiled tumor and stromal regions.”

4) *Abstract: “automated molecular histology approach” is cryptic and uninformative. This statement suggests the approach yielded images of 1000 proteoforms, which is misleading. Rephrase and clearly explain the intricacies of the approach resulting in different depth of coverage associated with different workflows.*

Response: We agree and therefore have removed the cryptic phrase and instead used a clear phrase to address the concern. *“We introduce AutoPiMS, a mass spectrometry (MS) imaging-based automated tandem MS workflow for proteoform identification directly from tissue microenvironments...From a total of ~1000 proteoforms detected by region-of-interest label-free quantitation, ...”*

5) *Lines 38-42: DDA MALDI-MSⁿ was previously demonstrated for lipids (<https://www.nature.com/articles/s41592-018-0010-6>) and could theoretically be accomplished for proteoforms in a similar fashion. Some more recent approaches (e.g., PASEF) have made huge strides for in-situ MSⁿ.*

Response: We thank the Reviewer for suggesting these techniques, and we would like to include these recent advances in the field in the introduction and references (Ref: 4-15, 19). However, we point out that MALDI predominantly generates singly-charged ions that significantly limits its ability to interface with top-down fragmentation due to the lack of sequence coverage in charged fragments. As the reviewer mentioned, novel MS² data acquisition approaches coupled to spatial and single-cell proteomics have made huge strides in the field bottom-up proteomics, where proteins are digested into peptides and are subjected to chromatographic/ion mobility separation prior to MS/MS fragmentation. However, whole proteoform information are typically not preserved in protein digestion steps in these workflows.

6) *Lines 42-43: Reads fragmented and does not introduce nanoDESI as written.*

Response: We have revised the sentence as (Line 45): *“When using electrospray ionization-based methods for proteoform desorption from tissues, each proteoform generates a distribution*

of charge states, resulting in congested spectra in the mass-to-charge (m/z) domain.” Moreover, we introduced nano-DESI in the second paragraph (Line 52).

7) Line 50: This is “nanoDESI” and not “DESI” (and these common abbreviations should be introduced). Or did the authors mean to reference FAIMS-DESI-MSI report where CID and UVPD were used to confirm several proteoforms up to ~20 kDa (<https://pubs.acs.org/doi/10.1021/acs.analchem.8b00967>)?

Response: We have changed DESI to nano-DESI according to the Reviewer’s suggestion. Moreover, we have revised the introduction to include all the techniques in the field of intact protein imaging and identification from tissues including the paper mentioned specifically by the reviewer in this comment (Line 50): “Recently, tissue spatial profiling and imaging with intact proteoform-level information has been demonstrated using techniques such as matrix-assisted laser desorption/ionization(MALDI), desorption electrospray ionization (DESI), nanospray desorption electrospray ionization (nano-DESI), picosecond infrared laser desorption by impulsive excitation (PIRL-DIVE), and nanodroplet processing in one pot for trace samples (NANOPOTS).”

8) Lines 51-55: This paragraph introduces I2MS, then launches directly into the AutoPiMS and LFQ, leaving the reader wondering as how AutoPiMS fits into the MSI workflow and connects with LFQ. What does “engine for molecular identification” refer to? The reader would benefit from breaking this down and clearly explaining the workflows employed and their connections and interdependencies.

Response: We have expanded this section of the manuscript to define AutoPiMS and explain its relationship with MSI. We have also revised the phrase “engine for molecular identification” (Line 62): “Herein, we developed “AutoPiMS”, a PiMS-based data-dependent MS² workflow for multiplexed proteoform identification directly off a tissue section ... AutoPiMS augments PiMS with a computational engine for unattended proteoform target selection and acquisition method generation...”

9) Lines 63-70: *Why are scan rates for the stages and MS2 speed stated here and then again in methods section? This led to some inconsistencies and should be avoided. (Also, no need to introduce abbreviation for Hertz.)*

Response: We have removed the scan rate description in the main text and removed the introduction of abbreviation of Hertz to avoid confusion as suggested by the Reviewer.

There are several different nanoDESI analyses included in this report, and AutoPiMS is only a part of the story. There should be a much clearer description of how these separate workflows play and integrate.

Response: We have reorganized the **Methods** section and reformulated the main text to describe all the different workflows we developed in this work, including AutoPiMS, PiMS imaging, and label-free quantitation. In particular, we have provided detailed responses to this question in Reviewer question 3).

10) Lines 101-109: *87 of 113 proteoforms, yet favorable windows only for 25? Are these 25 a subset of larger proteoforms within the dataset?*

Response: The Reviewer is correct. We have mentioned in Line 112 that “*We obtained 113 proteoform masses at >1% relative abundance ranging from 4-67 kDa from a single 40-minute survey line scan (Methods, Supplementary Fig. 2, and Supplementary Table 1), of which 25 were in the 17-50 kDa mass range (mass spectrum shown in Fig. 1b).*” To improve the readability of this section, we revised the sentence in Line 122: “*However, the process was able to identify favorable m/z windows for all 25 targets in the 17-50 kDa mass range...*”

11) *Supplementary figure 4 suggests these proteoforms were detected with incredibly low SNR (MS1), which is concerning. Please comment.*

Response: We reasoned that the Reviewer was referring to **Supplementary Fig. 5** in this comment. We note that the m/z spectra were obtained by reconstruction of the mass-domain spectra using absolute ion counts from detected proteoform in the mass domain, which does not necessarily show the true signal to noise of the detection. As described in the flagship I²MS publication (Kafader et al., *Nat. Methods*, 2020), I²MS is a single ion approach that every ion used to construct the spectrum has been stringently evaluated based on a variety of metrics (*e.g.*, STORI slope, R² value, time of birth, time of survival). As a result, conventional random noise does not typically appear in the mass-domain spectrum in I²MS. To avoid confusion and use this figure as an opportunity to demonstrate that the target proteoform is the dominant species in AutoPiMS selected m/z isolation windows from a complex proteoform mixture, we have replaced **Supplementary Fig. 5** by showing the absolute ion counts of all the proteoforms isolated in the selected m/z isolation windows in each case:

In each panel, proteoforms are listed from high to low in ion count. The red numbers in each panel indicate the absolute ion count of that proteoform in the isolation window. As we could see in this figure, the target proteoforms we intended to isolate were present with substantially higher ion count than other co-isolated proteoforms.

12) Line 110 (and elsewhere): Replace MS^2 spectra with tandem MS or fragment spectra

Response: As we define and use MS^1 and MS^2 throughout and these are accepted notation, we have left the acronyms in place and checked the consistency.

13) Line 118: Supplementary figure 6 seems to be out of context. It is unclear how this connects to regular MS^2 and I^2MS-MS^2 . What is the take home message?

Response: The take home message of **Supplementary Fig. 6** is that the AutoPiMS algorithm was able to spatially optimize the tandem MS acquisition events for 79 targets detected in a survey line scan. It belongs to AutoPiMS workflow for proteoform targets <17 kDa using regular ensemble MS^2 . We have revised the main text as follows to address this comment (Line 137): “Using AutoPiMS-embedded algorithms, we obtained favorable isolation conditions for 79 targets detected in the <17 kDa range in a representative survey line scan on the HGSOC tissue described in **Fig. 1a** (**Supplementary Fig. 6**).”

14) Line 119: 1% relative abundance of a noisy spectrum is not informative. SNR might be better performance metrics than relative abundance.

Response: As described in comment 10), second sub-response, the chemical and otherwise random noise present in conventional MS is not present in I^2MS in the same way. As such, the concept of SNR does not translate cleanly into the I^2MS space. Noise bands are not ubiquitous signals in I^2MS , and often proteoforms will be spaced by stretches of zero signal.

15) Lines 117-127: Clarify which instruments are being used. One could assume Q Exactive for the “ensemble mode” but was it the Plus or the HF? Provide some explanation as why so many different instruments have been used as this could potentially limit the applicability.

Response: We used Orbitrap Exploris 480 for ensemble mode AutoPiMS experiments. We have revised the text in this section to clarify the instrument platform used here (Line 135): “To

achieve a higher data acquisition rate for targets in <17 kDa range, we performed AutoPiMS workflow using Orbitrap Exploris 480 implemented with a 4 kV central electrode.”

Moreover, we have also revised the Methods section to clarify the instrument platform used for different experiments in this study (please see subsections of the “*AutoPiMS data acquisition*” section). We have also added a comment in the **Methods** section that I²MS mode AutoPiMS experiment may be performed on the Exploris system as well, which hopefully addresses the reviewer’s concern about the applicability of our approach.

16) What is meant by technical replicates in this context? Also, comment on reproducibly among replicates.

Response: Technical replicates refer to testing the AutoPiMS workflow using highly similar biological samples (adjacent tissue sections from the same patient tumor). Each of these replicates contain a different set of proteoforms depending on the outcome of the survey line scan, but all resulting in a high success rate in proteoform identification. We have revised the section as (Line 142): “*The same AutoPiMS workflow was repeated on two adjacent 10 μm sections of the tumor from the same patient, identifying 69% (18 out of 26) and 80% (16 out of 20) of the AutoPiMS-selected targets achieving an averaged success rate of >80% (Supplementary Tables 3, 4, and 5).*”

17) Line 152: Was AutoPiMS employed for LFQ? If so, how exactly was this accomplished? This information is not directly stated anywhere other than Figure 2a where it is unclear. This is important because of potential oversampling effects on downstream quantitation.

Response: AutoPiMS was employed for 17 highly differential proteoforms in tumor and stromal regions of the cancer tissue found from LFQ. However, AutoPiMS was not performed on the regions where LFQ was performed. We have clarified this point in the manuscript (Line 210): “*A survey line scan followed by an adjacent MS² scan were performed along the dashed line in Fig. 2a next to the LFQ-profiled tumor and stromal regions.*”

18) Figures are overly complicated and therefore difficult to follow and connect to explanations and conclusions provided in the text. Some of this material can be moved to SI to make the overall flow of the experiment clearer and connect better to the text. For example, Figures 1c and 1d could be moved to SI without losing any context and same goes for graphical fragment maps. In this case, less is more.

Response: We appreciate the Reviewer's suggestions about the figure complexity. However, we reasoned that these two panels demonstrate critical performance metrics of the novel data acquisition algorithms embedded in AutoPiMS and should stay in the main text. With the great suggestions from the Reviewer, we have put in substantial effort in the improving the flow of the revised manuscript to avoid confusions from multiple workflows and datatypes and keep AutoPiMS as the only focus in **Fig. 1**. We hope the Reviewer would appreciate the importance of these two figure panels while reading the revised manuscript.

19) Figure 2a, was AutoPiMS applied only along the vertical dashed line (and not implemented for the PiMS region)?

Response: Yes, the Reviewer is correct. We have also revised **Fig. 2a** and made the clarification in the manuscript (Line 210): "A survey line scan followed by an adjacent MS² scan were performed along the dashed line in **Fig. 2a**. next to the LFQ-profiled tumor and stromal regions."

20) How can one be certain that a monomethylation is present solely on Arg68 if only one AutoPiMS vertical line scan was applied?

Response: We agree with the Reviewer that strictly in terms of the supporting fragmentation in **Fig. 2f**, the monomethylation may not be localized to a single residue. Furthermore, in the stretch of six amino acid residues to which the monomethylation is localized, all six residues are viable candidates for monomethylation. However, in the literature and UniProt database PTM annotation (https://www.uniprot.org/uniprotkb/P50238/entry#ptm_processing, screenshot

below), arginine residues are more likely methylated compared to the other candidates, and the arginine residue is the only candidate in that region which can support dimethylation (*Clarke et al., Trends in Biochemical Sciences, 2013*), which was also detected and coarsely localized to the region. Therefore, the monomethylation can be confidently assigned to the arginine as the dominant site despite not being localized to single-residue precision through fragmentation data.

▶ Chain	PRO_0000075707	1-77	UniProt	Cysteine-rich protein 1
▶ Modified residue		9	UniProt	N6-acetyllysine ▶ Modified residue (large scale data)		12	PRIDE	Phosphotyrosine ▶ Modified residue		22	UniProt	N6-acetyllysine ▶ Modified residue (large scale data)		37	PRIDE	Phosphothreonine ▶ Modified residue (large scale data)		39	PRIDE	Phosphothreonine ▶ Modified residue (large scale data)		40	PRIDE	Phosphoserine ▶ Modified residue (large scale data)		57	PRIDE	Phosphotyrosine ▶ Modified residue		68	UniProt	Omega-N-methylarginine ▶ Modified residue (large scale data)		73	PRIDE	Phosphoserine 
To clarify this point, we have revised the manuscript accordingly (Line 225): “*The MS² data support the placement of the mono- and dimethylation on Arg68 as a major modification site*”.

21) Terminology is also confusing, “pixels” would suggest that the LFQ experiment could create images. If so, why not include these images?

Response: We have revised the terminology “pixels” as “sampled regions” in the sections describing LFQ experiment. As explained in Comment 3)-2, the goal of the LFQ experiment is to sample hundreds of samples containing from the histopathological region for quantitative interpretation of the proteoform detection level. Imaging dataset is best suited for reflecting the spatial abundance variations within the same region and does not align well with the experimental design of the LFQ experiments.

22) It is challenging to gauge small differences in the images with Jet or other color schemes, consider changing to viridis or ciridis. Overlaying of Jet with a red-scale image makes figure 2f hard to read.

Response: We agree with the Reviewer that the overlaid red image with a jet color scale causes confusions. We have changed the overlaid image to a two-channel image in **Fig. 2f** with VIM shown in blue and CRIP1 Arg68Me0 in red as follows:

23) *Figure 2 and related text: Ideally, the report should have included a validation of selected proteins (and/or PTMs) using complementary techniques.*

Response: We agree with the Reviewer that although we showed quantitative analysis, intact mass annotation and top-down tandem MS identification for each proteoform with an image, we did not include orthogonal validation in this manuscript, such as immunohistochemistry. However, we were able to include quantitation results of 8 proteins (**Supplementary Fig. 11**) we discussed in Figure 2 from a study of ovarian cancer tissues we have published previously (Hunt et al., *iScience*, 2021). In this previous study, tumor and stromal regions were sampled by laser capture microdissection and analyzed using TMT-LC-MS bottom-up proteomics workflow. 5 of the common proteins show consistent enrichment in tumor or stroma in these two studies. We have also revised the manuscript accordingly (Line 234): “*This explains the highly variable CRIP1 abundances in our previous bottom-up proteomics report on tumor- or stroma-enriched HGSOc samples (Supplementary Fig. 11).*”

24) *Just as the main body of the manuscript, the methods were hard to digest with three instruments with different settings used for different purposes. It is challenging to find the information with current organization (for example, identify parameters specific to the Q*

Exactive HF). One way to improve readability is to group all MS settings and relevant acquisition parameters into their respective “MSI”, “LFQ”, “AutoPiMS” section.

Response: We reasoned that this comment has been answered by the response to comment 3)-1. Specifically, we have revised and reorganized the **Methods** section to improve readability as the reviewer suggested here.

25) All supplementary tables are of high-quality, but it is unclear why is SNR reported only in selected tables.

Response: We reasoned that the Reviewer was referring to **Table S3**, the only table where S/N of the matching fragments were included. We have discussed and specifically pointed out in the **Methods** section that fragment spectra peak annotation employed a modified THRASH algorithm, which comes with a SNR value for each matched fragment in the mass-domain MS² spectrum. We did not employ the modified THRASH algorithm to pick peaks from other spectra shown in the Supplementary Information. Therefore, S/N values were not reported in other tables. The corresponding description in the **Methods** section is as follows: “*For MS² of >17 kDa proteoforms (P²MS datatype), a modified Thorough High Resolution Analysis of Spectra by Horn (THRASH) algorithm was applied to the spectra to determine fragment ion masses with a signal-to-noise cutoff of 3.*”

26) Supplementary Figure 2: Challenging to judge what 1% relative intensity corresponds to. Perhaps zooming on the lowest abundance would be enough to inform the reader? Specify which instrument was this acquired on, and which workflow was used: the LFQ on Exploris, MSI on Exploris or AutoPiMS demonstration survey scan on Q Exactive Plus (I2MS or normal). Reader shouldn't have to try to retrieve this information going back and forth between SI and the main body of the manuscript.

Response: We have revised **Supplementary Fig. 2** per the Reviewer’s request. Panel (b) was added to reflect the 100% and 1% abundance level, and the experimental settings were clarified in the caption.

27) Supplementary Figure 3: Same as above, which experiments do these spectra belong to?

Response: We have revised the caption as “*Reconstruction of m/z spectra of the same AutoPiMS survey line scan data in Supplementary Figure 2 ...*”.

28) Supplementary Figure 4: It is challenging to see any of the intensities in (b), and the plot is quite hard to see given the red coloring.

Response: We have revised **Supplementary Fig. 4** that we switched the color scale (green to dark-low to high) and used $\log_{10}(\text{Ion Count})$ scale to enhance the dominant low ion count data points:

29) Supplementary Figure 5: What is the SNR for these species? Does not seem to be above 5, and at 17% RA in (a), it does not seem to even be a peak? This is mildly concerning since it appears the algorithm isn’t picking the peaks.

Response: The Reviewer has precisely touched on the reason why this figure was constructed and displayed: to illustrate that m/z -based precursor selection is extremely limited. Fortunately, we use mass-based precursor selection instead. In this work, precursors were selected on a mass basis, which allowed for species of low abundance to be characterized. Selected masses are translated back into the m/z space for fragmentation, but as is seen in the figure, there might be significant overlap with other species in m/z space. This figure does not touch on our ability to use the spatial dimensions of the tissue to characterize targets in places where m/z overlap is minimal—which is certainly one way in which this issue is mitigated as described in the manuscript. However, if there is significant m/z overlap after accounting for these tools, the fragmentation spectra will undoubtedly contain signals corresponding to co-isolated proteoforms as opposed to the target of interest. These additional fragment signals are typically not incorrectly annotated as the target of interest, but they would lower confidence scores (*e.g.*, P-score) where applicable. We have made the intent of this approach clearer by revising the main text reference (Line 122): *However, the process was able to identify favorable m/z windows for all 25 targets in the 17-50 kDa mass range (Fig. 1c, Supplementary Table 2, and Supplementary Fig. 5) at or near their top abundance in space (Fig. 1d), even if they would have gone unannotated in a traditional m/z data-dependent acquisition mode.*

30) *Supplementary Figure 6: Which 79-targeted proteoforms are captured here and where do they come from, LFQ or MSI?*

Response: To address the Reviewer's question, we have revised the caption of **Supplementary Fig. 6** as "A heatmap of spatial relative abundances of 79-targeted proteoforms detected in the <17 kDa range in a representative survey line scan on the HGSOC tissue described in **Fig. 1a** (**Supplementary Table 6**)" and added the following text in the manuscript (Line 137): "Using AutoPiMS-embedded algorithms, we obtained favorable isolation conditions for 79 targets detected in the <17 kDa range in a representative survey line scan on the HGSOC tissue described in **Fig. 1a** (**Supplementary Fig. 6**)."

Reviewer 2

Excerpted General Comments:

This study developed a novel technique, AutoPiMS, for the detection of spatial proteoforms in human tissues. This technique is impressive due to its ability to combine potentially several methods into one, enabling the characterization of proteoforms in a 4-dimensional manner. Importantly the technique provides an opportunity to delineate (with accuracy and speed) proteoforms associated with tumour vs stroma cells with potential applications in many other solid tumours especially those in which the interplay between tumour and stroma cells have significance in therapy and patient outcome. The article is well written, detailed and with novel findings and as such should be published. One minor comment is for the authors to consider briefly highlighting the (potential) clinical significance of the differential expression of CRIP1 in tumour regions. Some studies (such as <https://www.ncbi.nlm.nih.gov/pmc/articles/PMC8820892/>) have alluded to this.

Response: We thank the reviewer for the positive comments, and focusing on the significance of the key results. We agree with the reviewer that the potential clinical significance of CRIP1 in cancer needs to be discussed in the manuscript. CRIP1 has been reported as a potential biomarker for a few cancer types. We have included a discussion as suggested by the reviewer and included a few citations (Line 238): “*Previous studies have reported elevated level of CRIP1 expression in gastric, prostate, and ovarian cancers.*”

Reviewer 3

Excerpted General Comments:

The significance of this study lies in the development of an integrated platform, AutoPiMS, that drives advancements in proteoform-level spatial biology. The platform enables four-dimensional characterization of proteoform signatures: intact molecular mass, spatial distribution, quantitative analysis of differential expression, and molecular identification. In particular, the

automated data acquisition engine enables proteoform identification up to ~54 kDa at a speed of <1 minute per proteoform directly off tissue. A spatially resolved study of ovarian cancer tissue with 20-micron resolution identified >300 differential proteoforms in stroma versus tumor regions from the same patient, including differential localization of methylated forms of CRIP1. This platform fills the gap between high-confidence proteoform discovery and spatial proteomics, opening up a new avenue for discoveries and precision diagnostics in clinical histology. I have a minor question. Have the researchers compared their findings to existing methods or performed any validation? It would be nice to have a specific comparison of the advantages and disadvantages of the autoPiMS method compared to traditional immunohistochemistry results.

Response: We thank the Reviewer for the positive comments. Because orthogonal validation of some findings by immunohistochemistry is challenging on limited specimens from the same patient samples after use by PiMS workflows, we focus on the technological advancements achieved using single ion counting. To address the Reviewer's concern, we were able to include quantitative results from other studies where 8 proteins we highlighted in **Fig. 2** from a study of ovarian cancer tissues we have published previously (Hunt et al., *iScience*, 2021; now included in **Supplementary Fig. 11**). In this previous study, tumor and stromal regions were sampled by laser capture microdissection and analyzed using TMT-LC-MS bottom-up proteomics workflow. Five of the common proteins show consistent enrichment in tumor or stroma in these two studies. We have also revised the manuscript accordingly (Line 217): *“This result is also consistent with previously published tumor- and stroma-enriched bottom-up proteomics study on HGSOB biopsies.”*

Summary: We hope that the revisions to the manuscript and our accompanying responses will help create an overall piece of sufficient excitement to make a compelling case for publication in *Nature Communications*. We are truly excited with this advance, and are forwarding it with the very sincere belief that it will be highly cited, readily adopted, and eventually lead to breakthroughs in technology. We view the ability to identify proteoforms directly off tissue using single ion mass spectrometry as a possible ‘game changer’ in the field of spatial proteomics. We look forward to hearing back about the next stage in the process.

With best regards,

Neil Kelleher

REVIEWERS' COMMENTS

Reviewer #1 (Remarks to the Author):

The authors have adequately addressed the comments and made appropriate revisions. They should carefully check and ensure the changes are implemented correctly and thoroughly for desired impact.

On behalf of our team, we are very grateful for the positive feedback from the peer reviewers. We are pleased to resubmit the revised manuscript, “*Automated imaging and identification of proteoforms directly from ovarian cancer tissue*”, for further consideration in *Nature Communications*.

We have adjusted the manuscript to reflect the reviewer’s recommendations and stand committed to publishing the clearest work possible that reaches the widest possible readership. Therefore, we present point-by-point responses (in plain, black text) to the comments from the reviewers (*highlighted in blue text and italicized*). We very much appreciate the reviewer’s dedication and attention to detail and believe their comments will elevate the quality of this work.

RESPONSE TO REVIEWERS' COMMENTS

Reviewer 1

Excerpted General Comments:

The authors have adequately addressed the comments and made appropriate revisions. They should carefully check and ensure the changes are implemented correctly and thoroughly for desired impact.

Response: We heartfully thank the reviewer for the positive comments, and for their time and care in evaluation of the work. We have carefully checked and ensure the changes are implemented correctly and thoroughly in the final production of the paper.

With best regards,

Neil Kelleher